# Coverage enhancement accelerates acidic CO$_2$ electrolysis at ampere-level current with high energy and carbon efficiencies

Xiaohan Yu[1,3], Yuting Xu[2,3], Le Li[1,3], Mingzhe Zhang[1], Wenhao Qin[1], Fanglin Che [2,4] ✉ & Miao Zhong [1,4] ✉

Acidic CO$_2$ electroreduction (CO$_2$R) using renewable electricity holds promise for high-efficiency generation of storable liquid chemicals with up to 100% CO$_2$ utilization. However, the strong parasitic hydrogen evolution reaction (HER) limits its selectivity and energy efficiency (EE), especially at ampere-level current densities. Here we present that enhancing CO$_2$R intermediate coverage on catalysts promotes CO$_2$R and concurrently suppresses HER. We identified and engineered robust Cu$_6$Sn$_5$ catalysts with strong *OCHO affinity and weak *H binding, achieving 91% Faradaic efficiency (FE) for formic acid (FA) production at 1.2 A cm$^{-2}$ and pH 1. Notably, the single-pass carbon efficiency reaches a new benchmark of 77.4% at 0.5 A cm$^{-2}$ over 300 hours. In situ electrochemical Fourier-transform infrared spectroscopy revealed Cu$_6$Sn$_5$ enhances *OCHO coverage ~2.8× compared to Sn at pH 1. Using a cation-free, solid-state-electrolyte-based membrane-electrode-assembly, we produce 0.36 M pure FA at 88% FE over 130 hours with a marked full-cell EE of 37%.

The electrochemical reduction of CO$_2$ (CO$_2$R) to valuable short-chain liquid feedstocks presents an elegant solution for storing intermittent renewable electricity, while also mitigating excessive CO$_2$ emissions resulting from the consumption of fossil fuels[1–3]. In recent decades, significant progress has been made in understanding CO$_2$R reaction pathways on catalysts and the associated energetics to control CO$_2$R selectivity, and in developing gas-diffusion-electrode-based membrane-electrode-assembly (MEA) electrolyzers capable of achieving industrially relevant CO$_2$R current densities over hundreds of mA cm$^{-2}$ [4–7]. Despite these efforts, further advancements in CO$_2$R technology will depend on improving full-cell energy efficiency (i.e., power-to-product efficiency) and maximizing CO$_2$ utilization efficiency (i.e., CO$_2$-to-product conversion).

Operating CO$_2$R in acidic electrolytes offers a potential approach to address significant CO$_2$ losses caused by the reaction between CO$_2$ and OH$^-$ in electrolytes to form CO$_3^{2-}$ during CO$_2$R in alkaline or neutral media. In strong acids with an electrolyte pH ≤ 1, carbonate formation

is rare[8]. However, the kinetics of CO$_2$R under such highly acidic conditions become retarded. Additionally, the substantial increase in surface *H coverage on catalyst surfaces causes a strong competing hydrogen evolution reaction (HER), resulting in a largely reduced Faradaic efficiency (FE) for CO$_2$R[9,10].

Early studies have shown that increasing the local concentrations of alkali metal cations in close proximity to catalyst surfaces can enhance CO$_2$R by stabilizing the surface-adsorbed CO$_2$ and CO$_2$R-related intermediates through non-covalent interactions, and, at the same time, impeding the diffusion of protons to the catalyst surfaces[11–13]. However, when the system operates at high potentials (i.e. high current densities), the electroconvective flows near the electrodes become turbulent and disordered. According to the Nernst-Planck equation, this turbulence can cause a fluctuation in cation concentrations near the electrode surface, leading to variations in the local electrical field that may have the potential to disrupt the beneficial cation effect on CO$_2$R[14]. Prior studies have shown that

[1]College of Engineering and Applied Sciences, Collaborative Innovation Center of Advanced Microstructures, National Laboratory of Solid State Microstructures, the Frontiers Science Center for Critical Earth Material Cycling, Nanjing University, Nanjing 210093, China. [2]Department of Chemical Engineering, University of Massachusetts Lowell, Lowell, MA 01854, USA. [3]These authors contributed equally: Xiaohan Yu, Yuting Xu, Le Li. [4]These authors jointly supervised this work: Fanglin Che, Miao Zhong. ✉e-mail: Fanglin_Che@uml.edu; miaozhong@nju.edu.cn

applying a micrometer-thick layer of nanoparticles as a surface coating layer on catalysts mitigates the irregularities in electro-convective flows within the coated microstructures, which helps maintain a steady concentration profile of alkali metal cations[15–17]. Despite this advancement in mitigating HER, the high local con-centration of alkali metal cations can lead to bicarbonate precipitation in the gas diffusion electrodes. The accumulation of bicarbonates hinders $CO_2$ diffusion and hampers $CO_2R$ efficiency. To address the bicarbonate precipitation challenge, strategies involve modifying catalysts with immobilized cation groups[18] or exploring $CO_2R$ in cation-free systems such as using solid-state-electrolyte (SSE) based MEA electrolyzers[19]. One priority for implementing these approaches is to develop efficient catalysts that can inherently pro-mote $CO_2R$ over the competing HER, particularly under strongly acidic conditions.

In this study, we employed computational investigations to dis-cover a $Cu_6Sn_5$ catalyst with a notable abundance of surface adsorp-tion sites having a large binding energy difference between *OCHO (the intermediate for formic acid (FA) production) and *COOH (the inter-mediate for CO production) while also having weaker binding to *H. This configuration allows for maximum selectivity in favor of produc-ing FA over CO and other hydrocarbons while simultaneously sup-pressing HER.

We experimentally constructed $Cu_6Sn_5$ alloy catalysts on poly-tetrafluoroethylene (PTFE) gas diffusion electrodes using thermal vapor deposition. In situ electrochemical attenuated total reflection Fourier-transform infrared (ATR-FTIR) spectroscopy analysis revealed a notable enhancement showcasing a ~2.8 elevation in *OCHO cover-age on $Cu_6Sn_5$ compared to control catalysts of Sn, operating under identical electrochemical conditions at pH = 1. The active $Cu_6Sn_5$ achieves a high FE of 91% for FA production at a current density of 1.2 A cm$^{-2}$ in a strongly acidic electrolyte at pH = 1. Furthermore, it shows a marked single-pass carbon efficiency (SPCE) of 77.4% at 0.5 A cm$^{-2}$, representing approximately 2.8-fold enhancement in the SPCE compared to the prior report of ~27.4% obtained at ~0.24 A cm$^{-2}$ [20]. Such notable performance was stable over 300 h of

continuous acidic $CO_2R$ at 0.5 A cm$^{-2}$ in a flow-cell electrolyzer at pH = 1.

When integrating $Cu_6Sn_5$ into a cation-free, SSE-based MEA elec-trolyzer, we produced 2.6 liters of 0.36 M pure FA at 100 mA cm$^{-2}$ (area: 4 cm$^2$) over 130 h at a production rate of 5 mL cm$^{-2}$ h$^{-1}$, along with a full-cell energy efficiency of 37%. These results feature the potential of enhancing surface intermediate coverage as an efficient means for acidic $CO_2R$, particularly in cation-free systems at indust-rially relevant current densities.

## Results

### Density functional theory (DFT) studies

Copper (Cu) has established itself as a predominant and cost-effective electrocatalyst capable of generating a variety of hydrocarbons through $CO_2R$. Previous research has revealed effective strategies for modifying the *CO-binding and *OCHO-binding properties on surfaces of Cu-based alloy surfaces by introducing foreign elements (e.g., Zn, Al, Pb) into the Cu lattice. Specifically, the inclusion of 5–10 at.% Zn or Al into Cu has shown the ability to partially weaken *CO adsorption on the Al or Zn modified Cu site compared to the adjacent Cu-Cu site, thereby creating asymmetric *CO binding energies for improved $C_{2+}$ production[16,17]. Additionally, the introduction of a single Pb atom into Cu has been observed to enhance selectivity towards formic acid[21]. These alternations selectively promote the production of CO/$C_{2+}$ or FA via distinctive reaction pathways (Fig. 1a). To further curb the com-peting HER in strong acids, two major approaches are considered: (1) employing alkali metal cations to hamper H$^+$ diffusion toward catalyst surfaces, and (2) augmenting the coverage of surface $CO_2R$ inter-mediates to outcompete the HER.

As a representative p-block metal, Sn has a relatively stronger oxygen affinity and weaker adsorption energy for *H than pristine Cu. It can also form alloys with Cu across the entire concentration profile ranging from 0% to 100%. Interestingly, a previous study has investi-gated the relationship between $CO_2R$-to-FA activity and the *OCHO adsorption energy with various metal catalysts[22], and identified Sn and Cu as promising candidates for selective FA production due to their

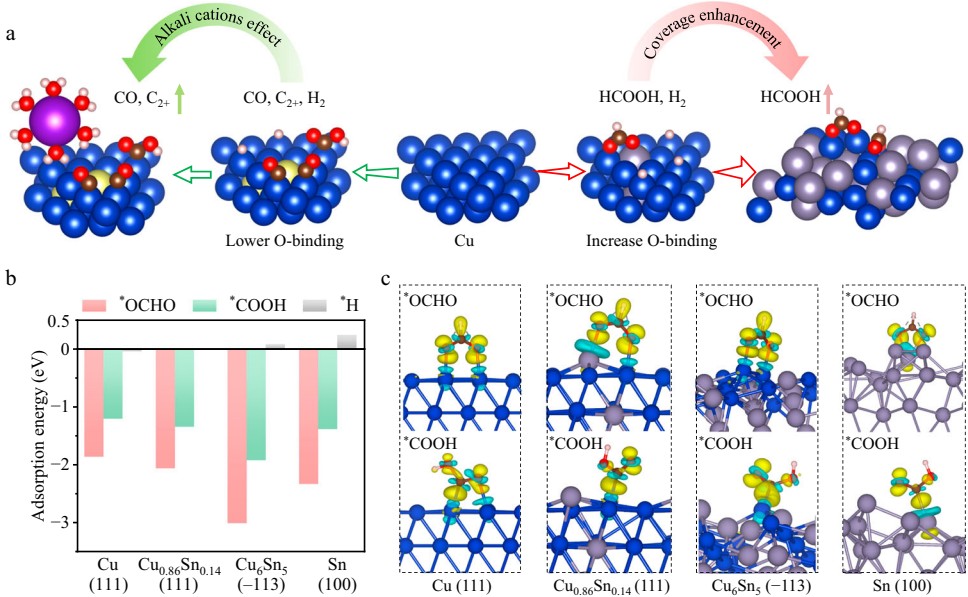

**Fig. 1 | DFT calculations. a** Schematic representation of $CO_2R$ and HER on Cu-based catalysts incorporated with different metal atoms, along with alkali cations effect and coverage enhancement strategy. Color-coded atoms represent Cu (blue), Sn (grey), Zn or Al (yellow), K (purple), O (red), C (brown), and H (pink). **b** The calculated adsorption energy of *OCHO, *COOH, and *H on Cu, $Cu_{1-x}Sn_x$ ($x$ = 0.14, 0.44), and Sn catalysts. Since the *H adsorption energies (referring to 1/2

$H_2$) over Cu and $Cu_{0.86}Sn_{0.14}$ are −0.03 eV and −0.02 eV, it is hard to distinguish their values in the plot. **c** Differential charge density plots illustrating the enhanced charge transfer between the $Cu_6Sn_5$ (−113) surface and *OCHO compared to the other examined surfaces. The isosurface level of the differential charge densities is 0.0045 e/bohr$^3$. The yellow or blue areas represent a gain or loss of electrons.

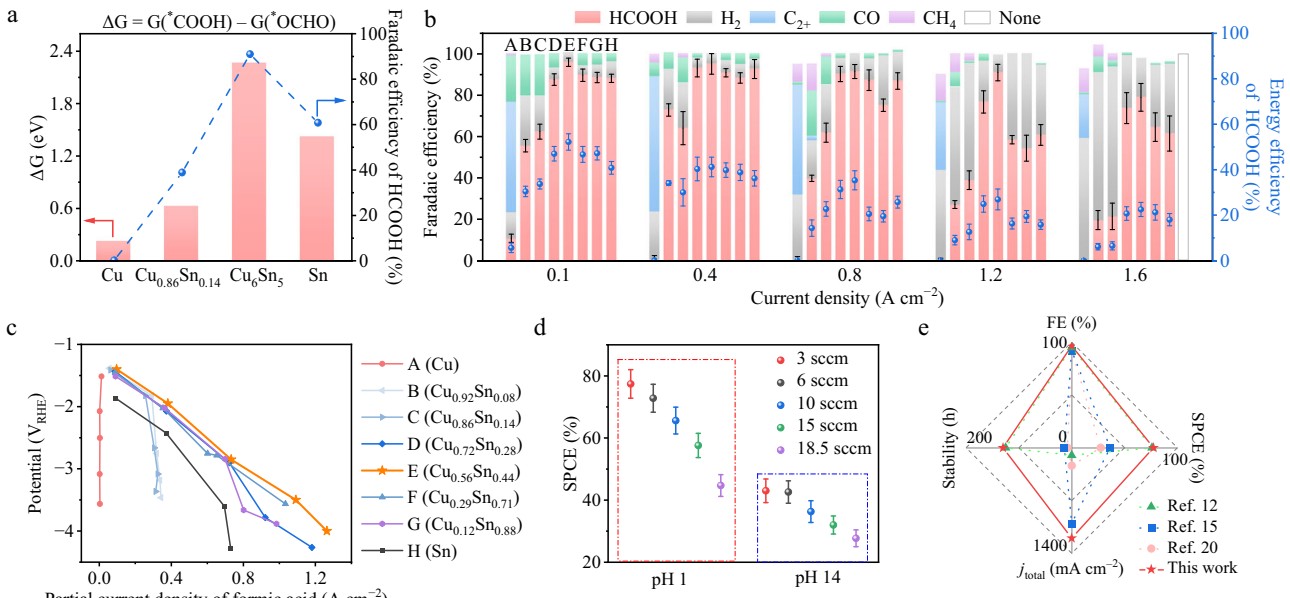

**Fig. 2 | Electrochemical performance of CO2 reduction with Cu, Cu$_{1-x}$Sn$_x$ ($x$ = 0.08, 0.14, 0.28, 0.44, 0.71, 0.88) and Sn catalysts in 3 M KCl and 0.05 M H$_2$SO$_4$ electrolyte at pH 1 in flow cells. a** A volcano-like plot showing the Gibbs free energy difference (ΔG) between ˙OCHO (an intermediate for FA production) and ˙COOH (an intermediate for CO production) over Cu (111), Cu$_{0.86}$Sn$_{0.14}$ (111), Cu$_6$Sn$_5$ (−113), and Sn (100) surfaces. This plot is in line with the obtained experimental data of Faradaic efficiency (FE) for FA production at 1.2 A cm$^{-2}$ in 3 M KCl and 0.05 M H$_2$SO$_4$. Gibbs free energy was calculated at room temperature (298.15 K), pH of 1, and an applied potential of −2.1 V vs. reversible hydrogen electrode (V$_{RHE}$) using computational hydrogen electrode (CHE) model. **b** Distributions of Faradaic

efficiencies (FE) for formic acid (FA), H$_2$, C$_{2+}$, CO, and CH$_4$ at 0.1, 0.4, 0.8, 1.2, and 1.6 A cm$^{-2}$. The error bars presented are derived from three independent tests. **c** Partial current densities of FA as a function of the applied potentials on Cu, Cu$_{1-x}$Sn$_x$ ($x$ = 0.08, 0.14, 0.28, 0.44, 0.71, 0.88), and Sn catalysts. **d** Single-pass carbon efficiency (SPCE) of CO2R on Cu$_6$Sn$_5$ at 0.5 A cm$^{-2}$, obtained at pH 1 and 14 at different CO$_2$ flow rates of 3, 6, 10, 15, 18.5 standard cubic centimeter per minute (sccm). The error bars presented are derived from three independent tests. **e** Performance comparison of this study and the previously published data under acidic conditions.

favorable ˙OCHO adsorption energies. With this insight, we aimed to create Cu-Sn alloys that could further fine-tune the ˙OCHO adsorption close to the optimal adsorption-energy values in the adsorption-activity trend. This would contribute to the improvement of the selectivity and activity in the CO$_2$R-to-FA conversion.

To understand the role of Sn in the Cu$_{1-x}$Sn$_x$ ($x$ = 0.14, 0.44) catalysts in tuning the catalytic selectivity of CO2R to CO and FA, we built surfaces with various Cu/Sn ratios from pure Cu to Cu$_{1-x}$Sn$_x$ alloy, and then to pure Sn, and used DFT to calculate the Gibbs free energetic diagrams for CO and FA generation. Based on the previous literature reviews[23,24], the potential rate-determining step (RDS) for the CO$_2$R-to-CO conversion is the first proton transfer to form ˙COOH from CO$_2$ (CO$_2$ (gas) + H$^+$ + e$^-$ + * → ˙COOH), while the potential RDS for CO$_2$R-to-FA is the first proton transfer to form ˙OCHO (CO$_2$ (gas) + H$^+$ + e$^-$ + * → ˙OCHO). By using DFT calculations, we identified that the Cu$_6$Sn$_5$ alloy possesses a high abundance of surface sites with a stronger binding affinity to ˙OCHO compared to ˙COOH, making it thermodynamically more favorable for producing FA (Fig. 1b). In addition, Cu$_6$Sn$_5$ exhibits notably weaker binding to ˙H (+ 0.33 eV referring to 1/2 H$_2$), resulting in a substantial energy requirement for H$_2$ production and thus, suppressing HER.

To study how the electronic properties of the surface-active sites on Cu$_{1-x}$Sn$_x$ catalysts with varying Sn concentrations affect the selectivity of FA and CO production, we calculated the differential charge density of ˙OCHO and ˙COOH over the experimentally observed surfaces in Fig. 1c. Our results indicate that the Cu$_6$Sn$_5$ (−113) surface exhibits more charge transfer with ˙OCHO than that over other examined surfaces. This charge analysis is consistent with the adsorption energy calculation, where the Cu$_6$Sn$_5$ (−113) surface exhibits the strongest adsorption energy of ˙OCHO among all the examined surfaces. Such strong adsorption energy of ˙OCHO also indicates a potentially high surface coverage of ˙OCHO over the

Cu$_6$Sn$_5$ (−113) surface. In addition, since two O−Cu bonds are formed between O$^{δ-}$ of ˙OCHO and Cu atoms on Cu$_{1-x}$Sn$_x$ surfaces, our results show that Cu$_6$Sn$_5$ (−113) surface presents -1.3 and -2.3 eV more favorable formation energy for ˙OCHO than for ˙COOH under alkaline (Supplementary Fig. 1) and acidic (Fig. 2a) conditions, respectively. Furthermore, according to the experimental results in Fig. 2b, the production of C$_{2+}$ and CH$_4$ products disappeared over the metal surface when the concentration of Sn in Cu$_{1-x}$Sn$_x$ catalysts was increased to -14%. To reveal the role of Sn in Cu$_{1-x}$Sn$_x$ catalysts in eliminating the C$_{2+}$ and CH$_4$ production during CO$_2$R, we compared the adsorption energy of CO over pure Cu (111) and Cu$_{0.86}$Sn$_{0.14}$ (111) surfaces (Supplementary Fig. 2). The DFT results show that the presence of Sn on the Cu$_{0.86}$Sn$_{0.14}$ (111) surface weakens the adsorption of CO to -0 eV compared to −0.08 eV on the pure Cu (111) surface. Thus, CO will likely desorb from the Cu$_{0.86}$Sn$_{0.14}$ (111) surface and be unable to hydrogenate via H protonation to produce CH$_4$ or couple with another adsorbed ˙CO intermediate to generate C$_{2+}$ species. These findings are in agreement with previous literature reports and experiments that have observed no C$_{2+}$ and CH$_4$ species produced over Cu$_{0.86}$Sn$_{0.14}$[25,26].

We performed Gibbs free energy calculations to quantitatively determine the reaction energy difference between the reaction pathways of CO$_2$-˙COOH−CO/C$_{2+}$ and CO$_2$-˙OCHO−HCOO$^-$ over the experimentally observed surfaces, including Cu (111), Cu$_{0.86}$Sn$_{0.14}$ (111), Cu$_6$Sn$_5$ (−113), and Sn (100) surfaces, where the possible reaction mechanisms have been shown in Supplementary Fig. 3. These calculated surfaces are the dominant facets in XRD measurement in Supplementary Figs. 4−9 and Supplementary Table 1. Gibbs free energetic diagrams for the formation of ˙OCHO and ˙COOH over different catalytic surfaces were constructed according to the experimental conditions of room temperature, pH 1 of the electrolyte, and an applied potential of −2.1 V vs. RHE (V$_{RHE}$). Our theoretical results (Fig. 2a,

Supplementary Figs. 10–33) show that the Gibbs free energy differences ($\Delta G = G(^*COOH) - G(^*OCHO)$) of the potential RDSs of FA and CO pathways during $CO_2R$ present a volcano-like plot as increasing the concentration of Sn in the $Cu_{1-x}Sn_x$ catalysts. Among the four examined surfaces, $Cu_6Sn_5$ (−113) shows the largest $\Delta G$ between the $CO_2R$ to FA and CO pathways, indicating that $Cu_6Sn_5$ (−113) contributes the highest selectivity for FA production. The theoretical results are consistent with the experimental results (Fig. 2b, c) that the FE of FA production over $Cu_6Sn_5$ is the highest.

To examine the competitive HER over Cu (111), $Cu_{0.86}Sn_{0.14}$ (111), $Cu_6Sn_5$ (−113) and Sn (100) surfaces, we examined the energy diagram of HER (Supplementary Fig. 34). The corresponding $^*H$ adsorption configurations can be seen in Supplementary Figs. 35–38. The increased free energy of HER indicates that the HER will be suppressed, and the selectivity for $CO_2R$ can thus be improved over $Cu_{0.86}Sn_{0.14}$, $Cu_6Sn_5$, and Sn compared to that on pristine Cu. This is consistent with experimental results (Fig. 2b), which show the FE of HER over Cu is higher than that on $Cu_{0.86}Sn_{0.14}$, $Cu_6Sn_5$, and Sn.

Finally, we built surfaces with different Cu/Sn ratios, ranging from pure Cu to $Cu_{1-x}Sn_x$ alloy, and then to pure Sn. Using DFT calculations, we analyzed the surface formation energetics for each surface. As depicted in Supplementary Fig. 39, $Cu_6Sn_5$ (−113) shows improved stability with a lower surface formation energy compared to the pristine Cu (111) facet. Conversely, Sn (100) shows poor stability with a higher surface formation energy.

## Electrochemical $CO_2R$ studies

With the theoretical insight, we conducted a systematical investigation into the Cu–Sn alloy as an illustrative catalyst system with the objective of obtaining exclusive $CO_2R$ to HCOOH at high current densities while also achieving high energy and carbon efficiencies. Experimentally, we created $Cu_{1-x}Sn_x$ ($x = 0.08, 0.14, 0.28, 0.44, 0.71, 0.88$) and control catalysts of pure Cu and Sn on PTFE gas diffusion electrodes using thermal evaporation (Supplementary Figs. 4 and 40, 41). We evaluated their $CO_2R$ performance for $Cu_{1-x}Sn_x$ ($x = 0.08, 0.14, 0.28, 0.44, 0.71, 0.88$) and controlled Cu and Sn catalysts in a flow cell with a three-electrode configuration in both alkaline and acid electrolytes. We quantitatively analyzed the gas and liquid products using gas chromatography (GC), nuclear magnetic resonance spectroscopy (NMR), and ion chromatography (IC).

In a 1 M KOH electrolyte at pH 14, the linear sweep voltammetry (LSV) curves in Supplementary Fig. 42 showed a sharp increase in the current densities with the increase of applied negative potentials, indicating efficient electrical conductivity for the Cu and $Cu_{1-x}Sn_x$ ($x < 0.71$) electrodes. With the addition of Cu in $Cu_{1-x}Sn_x$ ($x < 0.71$), the onset potential for formate shifted positively, indicating improved $CO_2R$ kinetics. At high current densities, $Cu_6Sn_5$ also displayed the highest selectivity for formate production, with over 90% formate FEs across a wide current density range of 0.4 to 1.2 A cm$^{-2}$ (Supplementary Fig. 43). The $Cu_6Sn_5$ catalyst showed formate partial current densities of up to -1.5 A cm$^{-2}$ at a relatively low operating potential of around −1.8 $V_{RHE}$ (Supplementary Fig. 44) and the highest formate selectivity close to 90% (Supplementary Fig. 43).

To assess the stability of $Cu_6Sn_5$ in 1 M KOH, we carried out Galvanostatic tests using an alternating current density mode (0.05 A cm$^{-2}$ for 30 s and 0.5 A cm$^{-2}$ for 90 s)[17] for 160 hours in a flow cell (Supplementary Fig. 45). Prior to the stability test, we coated a carbon-Nafion™ or SiC-Nafion™ mixed layer with a thickness of 2–5 μm on the surface of $Cu_6Sn_5$ to enable a uniform electrohydrodynamic flow near the catalysts[14]. Through the 120-h $CO_2R$ operation at a current density of 0.5 A cm$^{-2}$, we obtained a stable formate FE of above 85%. When comparing the FE and current density of this work with the previous reports under alkaline conditions, we found that we found that our results are superior (Supplementary Fig. 46 and Supplementary Table 2)[6,11,19,21,27–44].

In 3 M KCl and 0.05 M $H_2SO_4$ electrolyte at pH 1, $Cu_6Sn_5$ exhibited the highest selectivity, reaching up to 96% at −1.4 $V_{RHE}$ for formic acid in Fig. 2c. Remarkably, $Cu_6Sn_5$ showed a high FE of above 90% for FA production across a wide range of current densities from 0.4 to 1.2 A cm$^{-2}$ (Fig. 2b). We examined the SPCE for $Cu_6Sn_5$ under a constant current density of 0.5 A cm$^{-2}$ in a flow cell equipped with a 1.7 × 1.7 cm$^2$ serpentine channel reaction area at various $CO_2$ flow rates and electrolyte pH levels (more details are presented in the Supplementary Information)[15]. Figure 2d presents a maximum SPCE of 77.4%, achieved at a $CO_2$ flow rate of 3 standard cubic centimeter per minute (sccm) under pH 1 conditions. The electrode of $Cu_6Sn_5$ coated with carbon-Nafion™ or SiC-Nafion™ demonstrated stable FA production for over 300 hours, maintaining over 70% FE at a cathodic potential of −2.5 $V_{RHE}$. This equated to a 40% cathodic energy efficiency (CEE) at 0.5 A cm$^{-2}$ at pH 1 (Supplementary Fig. 47). Theoretical investigations were correspondingly carried out in this segment to investigate the effects of the electrochemical operating environment effects (e.g., electrolyte of 3 M K$^+$, applied potential of −2.1 $V_{RHE}$, and pH = 1) on the selectivity for FA vs. CO formation using Constant Electrode Potential (CEP) model via performing Grand canonical DFT calculations[45–49]. The selectivity for FA across Cu (111), $Cu_{0.86}Sn_{0.14}$ (111), $Cu_6Sn_5$ (−113), and Sn (100) surfaces exhibits a volcano plot (Fig. 2a and Supplementary Fig. 48), with $Cu_6Sn_5$ (−113) displaying the highest energy difference between the two pivotal intermediates, favoring the adsorption of $^*OCHO$ over CO. More details regarding grand canonical DFT simulations are provided in Supplementary Information. We compared this work with previously reported results on the $CO_2R$-to-FA production under acidic conditions in terms of elevated current density, FE, SPCE, and stability in Fig. 2e[12,15,20].

## Characterization

We conducted material and structure characterizations, as well as surface wettability analysis, for the $Cu_6Sn_5$ catalyst before the $CO_2R$ reaction. Figure 3a shows a schematic of the $CO_2R$ process on $Cu_6Sn_5$/PTFE electrodes. The cross-sectional scanning electron microscope (SEM) image in Fig. 3b shows the well-defined structure of $Cu_6Sn_5$ nano-to-micro particles densely packed on the PTFE substrate. This $Cu_6Sn_5$-on-PTFE structure was synthesized over a large scale via thermal evaporation (Fig. 3c and Supplementary Fig. 49), exhibiting hydrophobicity, as indicated by a water contact angle of 128° (Fig. 3c). This hydrophobicity enables $CO_2$ diffusion through the gaps between the particles on the PTFE sides to the catalyst surface (Fig. 3a). The top-view SEM presents that the diameter of the $Cu_6Sn_5$ particles is 300–600 nm (Fig. 3d). Energy dispersive X-ray spectroscopy in transmission electron microscopy (STEM-EDX) analysis revealed a uniform distribution of Sn and Cu over the majority of the $Cu_6Sn_5$ particles (Fig. 3e).

Detailed structural analysis using transmission electron microscope (TEM), high-resolution transmission electron microscope (HRTEM), and selective area electron diffraction (SAED) analyses (Fig. 3f, g) confirmed the formation of monoclinic $Cu_6Sn_5$ crystal. These results are in line with the X-ray powder diffraction (XRD) patterns in Supplementary Fig. 50, which indicate the presence of the main (−113), (132), and (−314) facets for $Cu_6Sn_5$. More details regarding the crystallographic and material analysis of catalysts after acidic $CO_2R$ operation are presented in Supplementary Figs. 51–53.

## In situ ATR-FTIR analysis

We used in situ ATR-FTIR spectroscopy to study $CO_2R$ intermediates coverage on $Cu_6Sn_5$-on-PTFE, Cu-on-PTFE, and Sn-on-PTFE electrodes during acidic $CO_2R$ electrolysis in the same pH 1 electrolyte of 3 M KCl and 0.05 M $H_2SO_4$, spanning a range of applied potentials from −0.24 to −1.64 $V_{RHE}$. We also incorporated an internal standard of potassium ferricyanide in the electrolytes to facilitate peak area calibration. In the

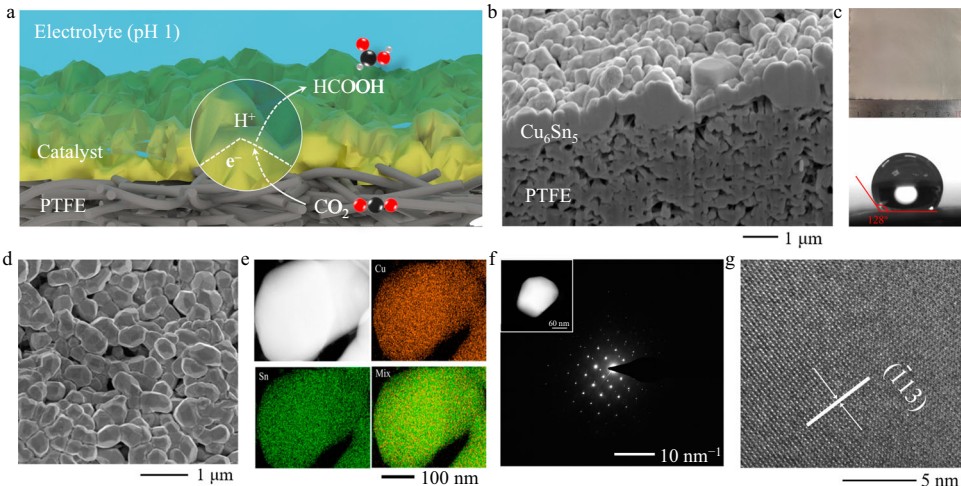

**Fig. 3 | Characterizations of the thermally evaporated Cu₆Sn₅-on-PTFE electrodes. a** Schematic representation of the CO₂R process on Cu₆Sn₅-on-PTFE electrodes. **b** Cross-sectional SEM image of the Cu₆Sn₅/PTFE electrode, showcasing the dense arrangement of Cu₆Sn₅ nano-to-micro particles on the PTFE substrate. **c** Optical image and the measured contact angle of a water droplet on the Cu₆Sn₅-on-PTFE electrode. **d**–**g** Top-view SEM image (**d**), STEM-EDX images (**e**), HAADF image and SAED pattern (**f**), and HRTEM image (**g**) of the thermally evaporated Cu₆Sn₅ catalyst.

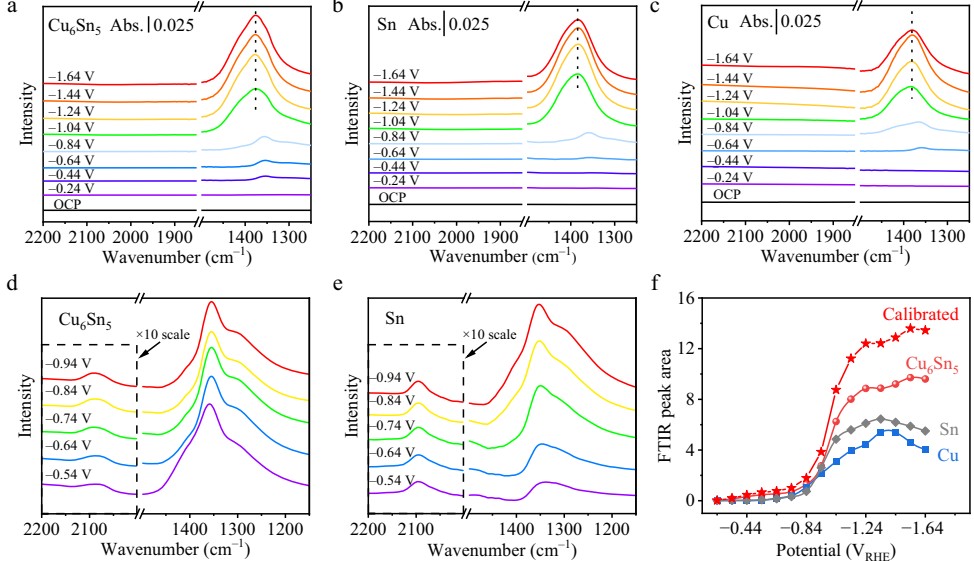

**Fig. 4 | In situ ATR-FTIR spectroscopy analysis. a**–**c** In situ ATR-FTIR spectra measured at different applied potentials for Cu₆Sn₅ (**a**), Sn (**b**), and Cu (**c**) under identical electrochemical conditions at pH = 1. Abs. stands for absorbance. **d**, **e** ATR-FTIR results of Cu₆Sn₅ (**d**), and Sn (**e**) in the potential range of −0.54 to −0.94 V$_{RHE}$, which were used to calculate the relative peak areas of ·OCHO using the peak areas of the potassium ferricyanide as a reference. The peak intensities of the internal standard potassium ferricyanide in panels (**d**) and (**e**) have been magnified by 10 times to facilitate comparison with those of the ·OCHO intermediates. **f** The integrated ATR-FTIR peak areas of the Cu, Sn, Cu₆Sn₅ with and without calibration.

ATR-FTIR spectra for the Cu₆Sn₅ and Sn catalysts (Fig. 4a–c), significant peaks were absent within the range of 1900 to 2200 cm$^{-1}$. In contrast, distinct infrared vibration bands related to the ·OCHO intermediate were observed within the 1375–1385 cm$^{-1}$ range for all Cu₆Sn₅, Cu, and Sn catalysts. The onset potential for the ·OCHO intermediate on Cu₆Sn₅ was −0.44 V$_{RHE}$, lower than that on Cu and Sn, indicating fast CO₂R-to-HCOOH kinetics with Cu₆Sn₅. During the in situ ATR-FTIR test using a 5 mM potassium ferricyanide internal standard in 3 M KCl and 0.05 M H₂SO₄, we observed a constant peak area for potassium ferricyanide within the potential range of −0.54 to −0.94 V$_{RHE}$. Consequently, we established a calibration curve within this potential range, using the potassium ferricyanide peak as a reference. This allowed us to calculate the peak areas of ·OCHO intermediates on different catalysts at the same potential range (Fig. 4d, e and Supplementary Table 3). Notably, adhering the established calibration relationship, the plateau ·OCHO peak intensity on Cu₆Sn₅ measured at potentials above −1.04 V$_{RHE}$ in the CO₂R electrolyte is ~2.8 times higher than that of Sn, with all measurements conducted under identical conditions (Fig. 4f). These results point toward an enhanced surface coverage of ·OCHO on Cu₆Sn₅, which facilitates selective FA production during acidic CO₂R.

To produce a pure formic acid solution, we conducted CO₂R using an AEM (Sustainion X37-50 Grade) based SSE-MEA electrolyzer. 0.5 M H₂SO₄ was used as the anolyte and IrOₓ/Ti foam was used as the anode (Fig. 5a). During the electrolysis, HCOO⁻ was generated on the cathode side and diffused through the AEM into the middle SSE layer.

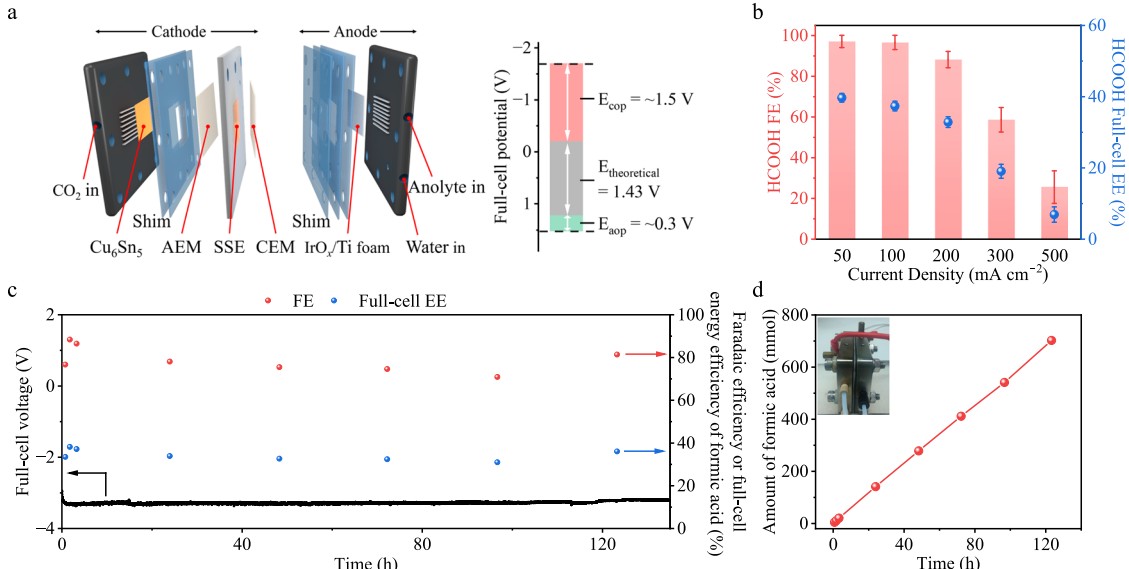

**Fig. 5 | Electrochemical CO₂R performance with Cu₆Sn₅ in acidic electrolytes at pH 1. a** Structure of a solid-state-electrolyte-based membrane electrode assembly electrolyzer for CO₂R and its potential distribution. **b** FEs and full-cell energy efficiencies of FA production with Cu₆Sn₅ at different current densities. The error bars presented are derived from three independent tests. **c** FEs and full-cell energy efficiencies of FA production with Cu₆Sn₅ at 100 mA cm⁻² during 130-h CO₂R operation. **d** Optical image of MEA and production rate of FA with Cu₆Sn₅ at 100 mA cm⁻² during 130-h CO₂R operation.

Simultaneously, $H^+$ permeated through the CEM (Nafion 117) and reacted with $HCOO^-$ to form FA. The produced FA solution was collected by passing deionized water through the SSE layer. As shown in Fig. 5a, the operating potential is 3.4 V for the overall $CO_2$-to-FA and $H_2O$-to-$O_2$ reaction in an SSE-based MEA electrolyzer. The detailed potential distributions are listed in Fig. 5a: a theoretical potential of 1.43 V required to initiate the $CO_2$R-to-FA reaction, a -0.3 V overpotential on the $IrO_x$ loaded Ti mesh anode, and an -1.5 V overpotential on the SSE, anion and cation membranes, and Cu₆Sn₅ cathode. In Fig. 5b, using the Cu₆Sn₅ catalyst, the FA FE reached ~96% at 100 mA cm⁻². The full-cell voltage was −3.7 V and the full-cell energy efficiency was over 37%. We evaluated the $CO_2$R performance in different anolytes (0.1 M, 0.5 M, and 1 M $H_2SO_4$) in Supplementary Fig. 54. We quantified the amount of FA produced using IC, NMR, and pH measurements (Supplementary Fig. 55). The results confirmed the production of 2.6 liters of 0.36 M pure FA solution (Supplementary Fig. 56) with a production rate of 5 mL cm⁻² h⁻¹ (20 mL h⁻¹) over a continuous 130-h $CO_2$R process (Fig. 5c). Figure 5d shows an optical image of an SSE-based MEA and a steady production rate of FA over the 130-h $CO_2$R process to produce ~700 mmol pure FA.

## Discussion

In this study, we demonstrate the effectiveness of enhancing surface coverage of $CO_2$R intermediates as a strategy to promote $CO_2$R at ampere-level current densities and under strongly acidic conditions. Theoretical DFT calculations have revealed that Cu₆Sn₅ enhances the adsorption of *OCHO compared to *COOH and *H, resulting in increased surface *OCHO coverage and promoted FA production in $CO_2$R while simultaneously suppressing the HER. Experiments have identified a robust Cu₆Sn₅ catalyst that exhibits exclusive FA production with over 90% FE at 1.2 A cm⁻², and a remarkable 77.4% carbon efficiency at 0.5 A cm⁻², maintaining stability over 300 h of continuous $CO_2$R operation at pH = 1. In situ electrochemical ATR-FTIR spectroscopy validates an approximately 2.8× enhancement in *OCHO coverage on Cu₆Sn₅ compared to the control catalyst of Sn, operating under identical electrochemical conditions at pH = 1. Furthermore, using an SSE-based MEA electrolyzer, we stably produce pure FA solution with a concentration of 0.36 M over a 130-hour reaction at a full-cell energy

efficiency surpassing 37%. We expect that the insights gained from our work, particularly in the context of optimizing intermediate adsorption and coverage, will provide valuable guidance for the advancement of selective and energy-efficient $CO_2$R with long-lasting performance in cation-free, MEA-based electrolyzers.

## Methods

### Density functional theory (DFT) calculation

DFT calculations were conducted using the Vienna Ab-initio Simulation Package (VASP)[50]. To take the solvation effects into consideration, hybrid solvation effects have been employed in VASPsol along with explicit water molecules at the interface[45–48]. The Constant Electrode Potential (CEP) model has been employed to investigate the constant negative applied potential and cation effects through grand-canonical DFT (GC-DFT) calculations[49]. These calculations were used to compare the energy diagrams of two possible $CO_2$ reduction reaction ($CO_2$R) pathways over different surfaces. We have investigated Cu (111), $Cu_{0.86}Sn_{0.14}$ (111), Cu₆Sn₅ (−113), and Sn (100) surfaces for calculations (Supplementary Fig. 4), where these facets are also the most thermodynamic favorable ones for Cu, Sn, and $Cu_{1−x}Sn_x$ alloy systems. More DFT calculation details are included in the Supplementary Information.

### Synthesis

We fabricated a series of $Cu_{1−x}Sn_x$ ($x = 0.08, 0.14, 0.28, 0.44, 0.71, 0.88$), Cu, and Sn electrocatalysts by thermal evaporation (SKY-RH400). Cu and Sn particles were placed separately in two Molybdenum boats inside the deposition chamber and melted slowly under the pressure of $10^{-5}$ Torr for thermal evaporation. We controlled the thermal evaporation rates of Cu and Sn to adjust the Cu/Sn ratios in the synthesized alloys. Evaporating rates of approximately $x$ Å s⁻¹ ($x = 0.08, 0.14, 0.28, 0.44, 0.71, 0.88$) for Sn and approximately $(1−x)$ Å s⁻¹ for Cu were used to produce alloyed $Cu_{1−x}Sn_x$ films on the polytetrafluoroethylene (PTFE) substrates. A quartz crystal monitor was used to observe the thicknesses of evaporated $Cu_{1−x}Sn_x$. Evaporation of pure Sn and pure Cu electrocatalysts was conducted using a similar fabrication procedure. The loading of all catalysts is 800 nm thick.

## Characterization

Transmission electron microscopy (SEM) images were taken using a Hitachi SU8100 SEM at an accelerating voltage of 5 kV. High-resolution transmission electron microscopy (HRTEM) and transmission electron microscopy-energy dispersive X-ray spectroscopy (TEM-EDX), selected area electron diffraction (SAED), and bright-field and dark-field TEM analyses were performed in a TEM (Tecn F20) with an accelerating voltage of 200 kV. X-ray powder diffraction (XRD) was carried out with a Bruker D8 Advance at a scanning rate of $10°$ $min^{-1}$ in the $2\theta$ range from $20°$ to $80°$. X-ray photoelectron spectroscopy (XPS) studies were performed using PHI5000 VersaProbe. The binding energy data were calibrated relative to the C 1 s signal at 284.6 eV.

## Electrochemical experiments

Experiments under alkaline and acidic conditions were performed in a flow cell using a three-electrode system. Ag/AgCl electrode was used as the reference electrode, commercial Ni foam (for use under alkaline conditions) and Pt (for use under acidic conditions) were used as the counter electrodes, $Cu_{1-x}Sn_x$ ($x = 0.08, 0.14, 0.28, 0.44, 0.71, 0.88$), Cu, and Sn on PTFE electrodes were used as the working electrodes (area: $0.5\,cm^2$), and the electrolytes were 1 M KOH (for alkaline $CO_2$ electrolysis) and 0.05 M $H_2SO_4$ with 3 M KCl (for acidic $CO_2$ electrolysis). Anion exchange membrane (Fumasep FAB-PK-130, size: $2 \times 2\,cm^2$, thickness: $130\,\mu m$) and proton exchange membrane (Nafion N117, size: $2 \times 2\,cm^2$, thickness: $183\,\mu m$) were used as the ion exchange membranes. The proton exchange membrane was immersed in 0.5 M $H_2SO_4$ for about 2 h before use. Experiments using solid-state electrolyte (SSE) were performed in a membrane electrode assembly (MEA) system with a 1 mm-thick SSE layer between the cathode and anode. $IrO_x$/ Ti foam was used as the anode, and $Cu_6Sn_5$ was used as the cathode (area: $4\,cm^2$). Anion exchange membrane (Sustainion X37-50 Grade, size: $4 \times 4\,cm^2$, thickness: $50\,\mu m$) and proton exchange membrane (Nafion N117, size: $4 \times 4\,cm^2$, thickness: $183\,\mu m$) were used as the ion exchange membranes. The SSE was AmberChrom 50WX4 hydrogen form (J&K Scientific), and the anolyte was 0.5 M $H_2SO_4$. The electrochemical workstation was Autolab PGSTAT302N (Metrohm). The gas flow rate was controlled at 25 mL $min^{-1}$ by an electronic flow meter, and the end flow rate was calibrated by a soap film flow meter during the test.

All of the electrode potentials vs. the Ag/AgCl electrode were converted to the potentials vs. reversible hydrogen electrode (RHE) using the following Eq. (1):

$$E_{RHE} = E_{Ag/AgCl} + 0.197 + 0.059 \times pH + iR \quad (1)$$

Gas-phase products were measured using gas chromatography (GC Agilent 990, Perkin Elmer Clarus 680). According to the peak area, the Faradaic efficiency of the gas products can be obtained, and the calculation formula is as the following Eq. (2):

$$FE = \frac{F \times z \times v \times n}{I \times t} \times 100\% \quad (2)$$

where $F$ is the Faraday's constant, which is 96485 C $mol^{-1}$. $z$ is the number of electrons required to reduce $CO_2$ to a CO or $H_2$ molecule, which is 2. $v$ is the gas flow rate, here is 25 mL $min^{-1}$. $n$ is the concentration of the gas products obtained by GC with 1 mL of sample gas, the unit is mol $mL^{-1}$. $I$ is the current applied to the sample, the unit is A. $t$ is the reaction time, and the unit is s.

Liquid product $HCOO^-$ was measured using ion chromatography (IC, SH-AC-11, Qingdao shenghan). The FE towards formate or FA at each current density was calculated by adding up both anodic and cathodic FEs. We first obtained a standard curve with a concentration gradient of sodium formate (HCOONa), $HCOO^-$ concentrations were set as 1 ppm, 2 ppm, 5 ppm, 10 ppm, 50 ppm, and 100 ppm (mg $L^{-1}$).

According to this, the $HCOO^-$ concentration in the electrolyte after the reaction can be obtained. The formula of faradaic efficiency calculation for $HCOO^-$ is as the following Eq. (3):

$$FE = \frac{F \times z \times V \times n}{45 \times I \times t} \times 100\% \quad (3)$$

where $F$ is the Faraday's constant, which is 96485 C $mol^{-1}$. $n$ is the concentration of $HCOO^-$ measured by the instrument based on the standard curve, and the unit is mg $L^{-1}$. $z$ is the number of electrons required for the reduction of $CO_2$ into $HCOO^-$, here is 2. $V$ is the volume of the catholyte, here is 0.03 L. $t$ is the reaction time, and the unit is s.

All data, including Faradaic efficiencies, were collected based on 1-hour electrolysis. Stability tests were conducted over 300 h. The overpotential is determined by subtracting the operating potential at specific current densities for formic acid generation from the theoretic potential for formic acid production, which is $-0.199\,V_{RHE}$. In situ electrochemical attenuated total reflection Fourier-transform infrared spectroscopy (ATR-FTIR) experiments were conducted on a Thermo Scientific Nicolet 6700 FTIR spectrometer with ZnSe as the prismatic window at room temperature. A three-electrode electrochemical single-cell was used for the tests (Supplementary Fig. 57). The thermally-evaporated $Cu_6Sn_5$ catalyst on a carbon gas diffusion layer (Freudenberg H15C13) was used as the working electrode, a Pt wire was used as the counter electrode, and a saturated Ag/AgCl electrode was used as the reference electrode. 3 M KCl and 0.05 M $H_2SO_4$ (pH = 1) saturated with $CO_2$ was used as the electrolyte. Open circuit potential (OCP) was conducted as a comparison, and the data was collected using chronoamperometric tests from $-0.24$ to $-1.64\,V_{RHE}$. The peak area of $^*OCHO$ in FTIR was calculated by integrating the corresponding curve areas in the same interval ($1250-1500\,cm^{-1}$) obtained at different potentials. More experimental details are discussed in the Supplementary Information.

## Data availability

Source data to generate figures and tables are available from the corresponding authors.

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

## Acknowledgements

The authors thank the National Natural Science Foundation of China (grant number 22272078), the National Key R&D Program of China (No. 2020YFA0406102), the Frontiers Science Center for Critical Earth Material Cycling of Nanjing University, the "Innovation and Entrepreneurship of Talents plan" of Jiangsu Province, and Program for Innovative Talents and Entrepreneurs in Jiangsu (JSSCTD202138). Y.X. and F.C. thank the sponsorship from the Department of Navy under the awards

N00014-22-1-2001 and N00014-20-1-2858 issued by the Office of Naval Research. The United States Government has a royalty-free license throughout the world for all copyrightable material contained herein. Y.X. and F.C. acknowledge the computational resources provided by the Massachusetts Green High-Performance Computing Center (MGHPCC), the computational resources provided by ACCESS Maximize Project No. CHM220016, ACCESS Accelerate Project No. CHE200083, and ACCESS Explore Project No. CHE220075.

## Author contributions

M.Z. conceived the idea and designed the experiments. X.Y., L.L., Ming.Z., and W.Q. conducted the synthesis, characterizations, and flow-cell tests. X.Y., L.L., and W.Q. designed and carried out the MEA tests. Y.X. and F.C. carried out DFT calculations and analysis. Y.X. and F.C. wrote the theoretical section of the manuscript. X.Y., L.L., Ming.Z., and M.Z. wrote the experimental and other sections of the manuscript. All authors discussed the results and contributed to the preparation of the manuscript. M.Z. and F.C. supervised the project.

## Competing interests

The authors declare no competing interests.
