## [Peer Review File · Nature Communications]

Coverage enhancement accelerates acidic CO₂ electrolysis at ampere-level current with high energy and carbon efficienciesREVIEWER COMMENTS

Reviewer #1 (Remarks to the Author):

In this manuscript, the authors reported a robust Cu₆Sn₅ catalysts for acidic CO₂ electroreduction to formic acid at ampere-level current. They found that the catalyst with diverse sites has strong *OCHO affinity and weak *H binding, enabling exclusive production of formic acid with high FE and current density in flow cells. This research is timely with multiple techniques of characterization and theoretical studies. Although the authors claim that increasing the surface coverage of intermediates on catalysts promotes CO₂R under acidic conditions, the focus of this manuscript seems to be predominantly on studies conducted under alkaline conditions. In addition, phase and structure modulation of bimetallic CuSn catalysts to HCOOH or CO are widely reported in alkaline and neutral media (ACS Catal. 2021, 11, 11103-11108; Appl. Catal. B-Environ. 2021, 292, 120119; Nat. Commun. 2018, 9, 4933; Nano Energy 2019, 59, 138-145; Chem. Commun. 2023, 59, 1054-1057; J. Mater. Chem. A 2019, 7, 27514-27521 etc.). The difference from the previous literature lies in the characterization of CuSn alloy performance for CO₂R under acidic conditions. Therefore, the authors should clarify the specificity of their catalyst, and explore in-depth mechanism of acidic CO₂R using CuSn catalysts. I think the manuscript is acceptable after major revision. Some comments are listed below:

- 1) As shown in Supplementary Figure 43, the XRD analysis indicates that the excessive Sn was completely removed after 5-10 hours of CO₂R reaction in alkaline (1M KOH) or acid (3M KCl and 0.05M H₂SO₄) electrolytes. Is Sn species stable in acidic electrolyte? The authors should confirm whether the Sn species on the surface of the catalyst was dissolved, while the Sn species in the bulk of the catalyst are retained after the reaction.
- 2) To confirm the formation of alloy, more characterizations and discussions are needed.
- 3) Different methods were utilized to evaluate electrochemical stability under alkaline (Supplementary Fig. 40) and acidic (Supplementary Fig. 45) conditions. The rationale for using different testing approaches should be provided, along with a comparison of which method is more appropriate.
- 4) The electrolyte environment significantly influences catalyst performance for CO₂ reduction, but such effects of different electrolytes were not considered in the theoretical calculations.
- 5) Since the electrolytic cell configuration substantially impacts CO₂R performance, testing consistency is needed. CO₂ performance was primarily evaluated in flow cells, whereas the in-situ ATR-FTIR tests were conducted in a three-electrode electrochemical single-cell. The rigor of the in-situ ATR-FTIR data is therefore questionable due to the mismatch in cell types.
- 6) Further verification is needed to confirm the assignment of the *OCHO intermediate in the ATR-FTIR spectroscopy analysis.
- 7) Infrared signal intensity depends on many factors. Direct comparison of peak area changes between different samples is insufficiently scientific. An internal standard methodology to benchmark relative peak areas may better verify variations in adsorption coverage. In addition, the approach for determining and comparing the peak areas in Fig. 4 is not explained. Details on the peak area calculation must be provided in the Methods.
- 8) To identify the active sites, it might be important to characterize the catalyst post-CO₂R more rigorously, such as morphology, oxidation states, and other physical and chemical information of the catalysts after stability testing.
- 9) The performance under different acidic conditions needs to be investigated, especially stronger acids like 0.1M or 1M H₂SO₄. Moreover, the practical applications of CO₂ reduction are hindered by the high concentration of K⁺ ions.
- 10) Improving the single-pass CO₂ utilization by reducing the CO₂ flow rate lacks scientific significance, since reducing the CO₂ flow rate may not actually enhance the catalytic activity. Providing the performance of CO₂R dependent on current density at different CO₂ flow rates is essential.

Reviewer #2 (Remarks to the Author):

The authors present a CuSn catalyst for high-rate CO₂ electrolysis in alkaline and acidic conditions, as well as in a solid-state electrolyte MEA to produce formic acid at competitive reaction rates. The selectivity was linked to the coverage of a key formic acid intermediate alongside weakened *H binding energy. Overall, the conclusions are well-supported and the work provides a significant step in the development of CO₂ electrolyzers. However, there is some key information missing and clarification on specific points required as mentioned below. Should the authors address these comments, I recommend publication of this manuscript in Nature Communications.

It is not clear how the overpotential for formic acid production was calculated. Is this the overpotential for a given current density, or the onset potential, in which case the generation of formic acid would need to be confirmed as a sole product. Can the authors please verify how this was assessed.

No data is provided for the other CuSn catalyst compositions which could be responsible for selectivity changes. It would be good to see this data to verify that the catalyst chosen was indeed the best-performing.

Information about the SPCE is missing. How long was each experiment to get a single point on the SPCE plot conducted for? What equations were used for SPCE calculations?

The authors mention that in pH < 1 solutions carbonate formation is rare, however recent reports rather show that it occurs close to the electrode but can be reconverted back to CO₂. The SPCE plots suggest that at pH 6.2, there is limited consumption of CO₂ by hydroxide to form carbonate. However, as the local pH increases under electrolysis conditions, the plots should display similar features to the alkaline feed. Can the authors explain this trend and account for the low loss of CO₂ to carbonate here.

Although XPS data is mentioned in the methods, no results are provided. There are required to assess the degree of electronic modulation that occurs upon alteration of the Sn/Cu ratio, as this data can also be correlated to selectivity changes due to the modulation of the d-band.

Electrolysis duration is not provided in most cases for data presented apart from for the long-term data. This key information is important to help verify that setups are not due to transient effects. This is particularly important for the SPCE plots.

Several studies are missing from figure S41 and the main text that show high FE for formate and should be added: <https://doi.org/10.1002/adv.201902989>;
<https://doi.org/10.1002/anie.202206279>; <https://doi.org/10.1002/anie.202110000>;
<https://doi.org/10.1002/adma.202002822>; DOI: 10.1002/adfm.202213145

The authors should provide potentials for a range of current densities to facilitate comparison of energy efficiency with other work.

Line 113 – the reference to Figure 1a is not described in detail. The authors should provide a clear explanation to the altered pathways available when heteroatoms are present or simplify the schematic to clearly show the different pathways.

In Fig S45, what was the flow rate of CO₂ in this experiment and the corresponding SPCE?

Line 268 – I could not find the additional crystallographic data mentioned.

Response to Reviewers

Reviewer #1 (Remarks to the Author):

In this manuscript, the authors reported a robust Cu₆Sn₅ catalysts for acidic CO₂ electroreduction to formic acid at ampere-level current. They found that the catalyst with diverse sites has strong *OCHO affinity and weak *H binding, enabling exclusive production of formic acid with high FE and current density in flow cells. This research is timely with multiple techniques of characterization and theoretical studies. Although the authors claim that increasing the surface coverage of intermediates on catalysts promotes CO₂R under acidic conditions, the focus of this manuscript seems to be predominantly on studies conducted under alkaline conditions. In addition, phase and structure modulation of bimetallic CuSn catalysts to HCOOH or CO are widely reported in alkaline and neutral media (ACS Catal. 2021, 11, 11103-11108; Appl. Catal. B-Environ. 2021, 292, 120119; Nat. Commun. 2018, 9, 4933; Nano Energy 2019, 59, 138-145; Chem. Commun. 2023, 59, 1054-1057; J. Mater. Chem. A 2019, 7, 27514-27521 etc.). The difference from the previous literature lies in the characterization of CuSn alloy performance for CO₂R under acidic conditions. Therefore, the authors should clarify the specificity of their catalyst, and explore in-depth mechanism of acidic CO₂R using CuSn catalysts. I think the manuscript is acceptable after major revision. Some comments are listed below:

Response: We express our gratitude to the reviewer for approving our work and providing valuable comments that have enabled us to enhance the quality of our work. Our detailed actions and responses have been documented in the point-by-point response below, and we have highlighted the changes in yellow in the revised manuscript.

We also agree with the reviewer's point regarding the focus of this work, which centers on identifying a robust Cu₆Sn₅ catalyst for high-performance acidic CO₂R. Consequently, we have conducted a series of new performance and characterization studies for CO₂R with Cu₆Sn₅ and control catalysts under acidic conditions. Specifically, we compared the acidic CO₂R performance for all Cu_{1-x}Sn_x ($x = 0.08, 0.14, 0.28, 0.44, 0.71, 0.88$) and control catalysts of pure Cu and Sn in 3 M KCl and 0.05 M H₂SO₄ electrolytes at pH = 1 in flow cells. Figure R1 illustrates that Cu₆Sn₅ exhibits the highest Faradaic efficiencies for formic acid production across a broad range of applied current densities from 0.1–1.6 A cm⁻². This result aligns with our computational results (Figure 1 in the manuscript) and *in situ* ATR-FTIR findings (Figure 4 in the manuscript), affirming that an increased surface *OCHO coverage promotes formic acid production. Figures R2 and R3 depict the characterization results for the Cu₆Sn₅ catalyst after the acidic CO₂R, and Figure R4 describes the computational results regarding the surface formation energies of Cu (111), Cu_{0.86}Sn_{0.14} (111), Cu₆Sn₅ (-113) and Sn (100) surfaces. These results further confirm the stability of Cu₆Sn₅ catalyst.

All in all, in light of the reviewer's recommendations, we have revised Figure 2 in the manuscript to incorporate the acidic CO₂R data and moved the alkaline CO₂R data to Figure S43 in the revised Supplementary Information.

Figure R1. Faradaic efficiencies (FEs) and energy efficiencies (EEs) for Cu, Cu_{1-x}Sn_x ($x = 0.08, 0.14, 0.28, 0.44, 0.71, 0.88$) and Sn catalysts at various current densities of 0.1, 0.4, 0.8, 1.2, and 1.6 A cm⁻² in 3 M KCl and 0.05 M H₂SO₄ at pH = 1 in flow cells. The error bars presented are derived from three independent tests.

Figure R2. Comparison of X-ray diffraction (XRD) patterns for Cu₆Sn₅ before and after the 120-hour stability test at 0.5 A cm⁻² under both pH 1 and pH 14 conditions.

Figure R3. (a) Scanning electron microscopy (SEM), (b) high-resolution transmission electron microscopy (HRTEM), and (c–h) scanning transmission electron microscopy with energy dispersive X-ray spectroscopy (STEM-EDX) images of Cu_6Sn_5 following the 120-hour stability test conducted at 0.5 A cm^{-2} at pH 1.

Figure R4. DFT calculated surface formation energies for $\text{Cu}_{0.86}\text{Sn}_{0.14}$ (111), Cu_6Sn_5 (-113), and Sn (100) surfaces compared with Cu (111) surface where Cu (111) surface formation energy is $\sim 0.98 \text{ J/m}^2$.

- 1) As shown in Supplementary Figure 43, the XRD analysis indicates that the excessive Sn was completely removed after 5-10 hours of CO_2R reaction in alkaline (1M KOH) or acid (3M KCl and 0.05M H_2SO_4) electrolytes. Is Sn species stable in acidic electrolyte? The authors should confirm whether the Sn species on the surface of the catalyst was dissolved, while the Sn species in the bulk of the catalyst are retained after the reaction.

Response: Thank you for the comment. Our experiments have indeed shown that the evaporated excessive Sn on Cu_6Sn_5 was partially removed after 5–10 hours of alkaline CO_2R in 1 M KOH, while the Cu_6Sn_5 catalyst remained stable throughout an extended CO_2R operation spanning over 120 hours. This result is confirmed by

the X-ray diffraction (XRD) patterns for Cu₆Sn₅ before and after alkaline CO₂R, as depicted in Figure S43. In response to the reviewer's recommendations, we conducted CO₂R for Cu₆Sn₅ in a 3 M KCl and 0.05 M H₂SO₄ electrolyte at pH 1 for 120 hours. Prior to the acidic CO₂R, we coated a SiC-Nafion protection layer on the Cu₆Sn₅ surfaces to enhance the catalyst stability. Notably, stable formic acid production with 70–80% Faradaic efficiencies was achieved during the 120-hour acidic CO₂R. Subsequent XRD analysis of the post-reaction Cu₆Sn₅ electrode revealed that the evaporated excessive Sn was partially removed, while Cu₆Sn₅ remain well maintained (Figure R2). Further validation through scanning electron microscopy (SEM), high-resolution transmission electron microscopy (HRTEM), and scanning transmission electron microscopy with energy dispersive X-ray spectroscopy (STEM-EDS) mapping confirmed the morphology and crystallinity for Cu₆Sn₅ in the micro-structures (Figure R3). Specifically, the active Cu₆Sn₅ (–113) facet was clearly observed in HRTEM and XRD patterns in the post-reaction sample. These experimental results support our conclusions that Cu₆Sn₅ remains stable during long-term CO₂R under both acid and alkaline conditions.

Computational studies of surface formation energies of various examined catalytic surfaces were conducted to compare the stability of these catalysts, including Cu (111), Sn (100), Cu_{0.86}Sn_{0.14} (111), and Cu₆Sn₅ (–113). Figure R4 illustrates the DFT calculations pertaining to the surface formation energies of various surfaces. The surface formation energy can be calculated with following equations (Ref. 5 in the revised Supplementary Information):

$$E_{sur} = \frac{E_{total}}{2A} = \frac{E_{slab} - N \cdot E_{bulk}}{2A} \quad (R1)$$

$$E_{sur} = \frac{E_{total}}{2A} = \frac{E_{slab} - N_{Cu} \cdot E_{Cu-bulk} - N_{Sn} \cdot E_{Sn-bulk}}{2A} \quad (R2)$$

where pure Cu, Sn, and Cu₆Sn₅ (–113) surfaces can be calculated with Equation (R1). E_{slab} and E_{bulk} denote the energy of the total surfaces and bulk energy of Cu in the FCC structure, Sn in the tetragonal structure, and Cu₆Sn₅ alloy in monoclinic structure, (Ref. 6, 7 and 8 in the revised Supplementary Information) respectively. A is the surface area and N is the number of atoms in the surfaces. The surface formation energy of Cu_{0.86}Sn_{0.14} (111) can be calculated with Equation (R2). E_{Cubulk} and E_{Snbulk} denote the bulk energy of Cu and Sn, respectively. The Cu_{0.86}Sn_{0.14} (111) surface corresponds to the Cu (111) surface with Sn atoms introduced as dopants. The values for E_{Cubulk} and E_{Snbulk} in this context are derived from calculations based on pure Cu and Sn bulk materials. Our results on the surface formation energy of Cu (111) agree well with previous investigations, in which the surface formation energy of the Cu (111) surface in our study is determined to be 0.98 J/m² and the corresponding value in the open literature derived from DFT calculations is reported to be 1.17 J/m² (Ref. 9 in the revised Supplementary Information).

As shown in Figure R4, compared to the surface formation energy of Cu (111), it is evident that the Cu₆Sn₅ (–113) surface is more stable with a notably lower surface formation energy. While the Sn (100) surface is

less stable with a higher surface formation energy than that of Cu (111). These results agree well with the experiments regarding the great stability of the Cu_6Sn_5 catalyst during prolonged acidic CO_2R .

We have incorporated these new experimental and theoretical studies, along with relevant discussions, in the revised manuscript in lines 192 to 197, on page 8, and in the revised Supplementary Information in lines 124 to 144, and in Figure S39, on pages 5–6.

Figure R2. Comparison of X-ray diffraction (XRD) patterns for Cu_6Sn_5 before and after the 120-hour stability test at 0.5 A cm^{-2} under both pH 1 and pH 14 conditions.

Figure R3. (a) Scanning electron microscopy (SEM), (b) high-resolution transmission electron microscopy (HRTEM), and (c–h) scanning transmission electron microscopy with energy dispersive X-ray spectroscopy (STEM-EDX) images of Cu_6Sn_5 following the 120-hour stability test conducted at 0.5 A cm^{-2} at pH 1.

Figure R4. DFT calculated surface formation energies for Cu_{0.86}Sn_{0.14} (111), Cu₆Sn₅ (-113), and Sn (100) surfaces compared with the Cu (111) surface where Cu (111) surface formation energy is ~0.98 J/m².

The manuscript has been revised on Page 8:

We built surfaces with different Cu/Sn ratios, ranging from pure Cu to Cu_{1-x}Sn_x alloy, and then to pure Sn. Using DFT calculations, we analyzed the surface formation energetics for each surface. As depicted in Supplementary Fig. 39, Cu₆Sn₅ (-113) shows improved stability with a lower surface formation energy compared to the pristine Cu (111) facet. Conversely, Sn (100) shows poor stability with a higher surface formation energy.

The Pages 5–6 in Supplementary Information part have been revised as following for the DFT calculations:

For the surfaces mentioned above, the surface formation energy can be calculated with following equations⁵:

$$E_{sur} = \frac{E_{total}}{2A} = \frac{E_{slab} \cdot N \cdot E_{bulk}}{2A} \quad (7)$$

$$E_{sur} = \frac{E_{total}}{2A} = \frac{E_{slab} \cdot N_{Cu} \cdot E_{Cu-bulk} + N_{Sn} \cdot E_{Sn-bulk}}{2A} \quad (8)$$

where pure Cu, Sn, and Cu₆Sn₅ (-113) surfaces can be calculated with Equation (7). E_{slab} and E_{bulk} denote the energy of the total surfaces and bulk energy of Cu in a face centered cubic (FCC) structure, Sn in the tetragonal structure and Cu₆Sn₅ alloy in monoclinic structure,^{6,7,8} respectively. A is the surface area and N is the number of atoms in the surfaces. The surface formation energy of Cu-Sn alloy surfaces can be calculated with Equation (8). E_{Cubulk} and E_{Snbulk} denote the bulk energy of Cu and Sn atoms, respectively. The Cu_{0.86}Sn_{0.14} (111) surface corresponds to the Cu (111) surface with Sn atoms introduced as dopants. The values for E_{Cubulk} and E_{Snbulk} in this context are derived from calculations based on pure Cu and Sn bulk materials. Our results on the surface formation energy of Cu (111) agree well with previous investigations, in which the surface formation energy of the Cu (111) surface in our study is determined to be 0.98 J/m² and the corresponding value in the open literature derived from DFT calculations is reported to be 1.17 J/m².⁹

As shown in Supplementary Fig. 39, compared to the surface formation energy of Cu (111), it is evident that the Cu_6Sn_5 (-113) surface is more stable with a notably low surface formation energy. While the Sn (100) surface is less stable with a higher surface formation energy than that of Cu (111). These results agree well with the experiments regarding the great stability of the Cu_6Sn_5 catalyst and the poor stability of the Sn catalyst during prolonged acidic CO_2R .

Supplementary Fig. 39 DFT calculated surface formation energies for $\text{Cu}_{0.86}\text{Sn}_{0.14}$ (111), Cu_6Sn_5 (-113), and Sn (100) surfaces compared with Cu (111) surface where Cu (111) surface formation energy is $\sim 0.98 \text{ J/m}^2$.

- 2) To confirm the formation of alloy, more characterizations and discussions are needed.

Response: Thank you for the comment. We have performed additional characterizations including HRTEM, SEM-EDS, and XRD analyses for Cu_6Sn_5 after the 120-h acidic CO_2R . Our examination of the crystallinity data, both before and after the reaction, suggest a stable Cu_6Sn_5 catalyst for CO_2R .

Specifically, the XRD results for the evaporated samples before and after both acid and alkaline CO_2R revealed distinct peaks corresponding to the Cu_6Sn_5 phase. The relative intensities of these XRD peaks align with the Cu_6Sn_5 reference data (Figures R2 and R3). HRTEM and SEM-EDS results for Cu_6Sn_5 before and after CO_2R both illustrated a clear Cu_6Sn_5 (-113) plane, with an approximately 6:5 atomic ratio of Cu to Sn (Figures R3, R5 and Figure 3g in the manuscript). Considering that XRD gives crystal information on a macro millimeter scale and HRTEM on a micro nanometer scale, we suggest that Cu_6Sn_5 was formed on PTFE substrates over a large area via thermal evaporation and remained intact after acidic CO_2R . We have presented a more detailed discussion of these findings on page 12 of the revised manuscript and pages 61–63 of the revised Supplementary Information.

Figure R2. Comparison of X-ray diffraction (XRD) patterns for Cu_6Sn_5 before and after the 120-hour stability test at 0.5 A cm^{-2} under both pH 1 and pH 14 conditions.

Figure R3. (a) Scanning electron microscopy (SEM), (b) high-resolution transmission electron microscopy (HRTEM), and (c–h) scanning transmission electron microscopy with energy dispersive X-ray spectroscopy (STEM-EDX) images of Cu_6Sn_5 following the 120-hour stability test conducted at 0.5 A cm^{-2} at pH 1.

Figure R5. Scanning electron microscopy with energy dispersive X-ray spectroscopy (SEM-EDX) images of Cu_6Sn_5 following the 120-hour stability test conducted at 0.5 A cm^{-2} at pH 1.

- 3) Different methods were utilized to evaluate electrochemical stability under alkaline (Supplementary Fig. 40) and acidic (Supplementary Fig. 45) conditions. The rationale for using different testing approaches should be provided, along with a comparison of which method is more appropriate.

Response: Thank you for the comment. We have now provided a more detailed discussion on the differences in testing conditions for alkaline and acidic CO_2R on page 58 in the revised Supplementary Information:

- In the case of alkaline CO_2R , we employed an alternating current density mode (0.05 A cm^{-2} for 30 s and 0.5 A cm^{-2} for 90 s) to assess CO_2R performance. This alternative current density mode, as previously reported in (*ACS Energy Lett.* 2021, **6**, (2), 809–815; *ACS Catal.* 2020, **10**, 12403–12413), mitigates carbonate precipitation during alkaline CO_2R , ensuring the stable diffusion of CO_2 for continuous CO_2R .
- In the case of acidic CO_2R , the use of strong acid electrolytes mitigates severe carbonate precipitation. Therefore, we adopted normal chronopotentiometry to evaluate CO_2R performance under these conditions.
- 4) The electrolyte environment significantly influences catalyst performance for CO_2 reduction, but such effects of different electrolytes were not considered in the theoretical calculations.

Response: Thank the reviewer for the suggestion. We performed additional DFT calculations with considering

the electrolyte and applied potential effects on the CO₂R-to-formic acid (FA) and CO₂R-to-CO.

Here, we applied the hybrid solvation effects using constant electrode potential (CEP) model via performing grand-canonical DFT calculations (GC-DFT). We compared the formation energy of the two crucial intermediates (*OCHO and *COOH) to represent the selectivity of CO₂R to formic acid vs. CO. To account for solvation effects, we employed both the implicit solvation model incorporated in VASPsol, along with explicit water molecules at the interface (Ref. 18–21 in the revised Supplementary Information). In the VASPsol model, the solvent was treated as a continuous medium, and its impact was represented using the dielectric constant of water. The electrolyte solution was modeled using a linearized Poisson-Boltzmann approach. Additionally, we implemented the Constant Electrode Potential (CEP) model to investigate surface charge and cation effects (e.g., K⁺) on the CO₂R reaction through GC-DFT calculations (Ref. 22 in the revised Supplementary Information). At room temperature, the Debye screening length (Å) was calculated from the electrolyte concentration (M) using the following equation (Ref. 23 in the revised Supplementary Information):

$$\kappa = \frac{3}{\sqrt{I}} \quad (\text{R3})$$

In the equation, κ represents the Debye length, and I denotes the electrolyte concentration (e.g., 3 M K⁺) in this part. In the CEP model, the applied potential was adjusted the Fermi level (E_f) towards a desired value by manipulating the number of electrons added or removed within the system. This adjustment by various number of the electrons can tune the work function (Φ) of the system and, thus, tune the applied potential (U_{SHE}) with a reference of the work function of Standard Hydrogen Electrode (Φ_{SHE}):

$$U_{SHE} = \frac{\Phi - \Phi_{SHE}}{e} \quad (\text{R4})$$

In the given context, Φ_{SHE} represents the work function of the standard hydrogen electrode, determined as $\Phi_{SHE} = 4.43$ eV using RPBE for the thermodynamic work function of the standard hydrogen electrode (Ref. 24 in the revised Supplementary Information). This value serves as a reference for adjusting the number of additional electrons introduced into the systems. By altering the number of extra electrons added into the systems, we ensure a consistent potential of ~ -2.1 V vs. RHE for each examined elementary reaction, aligning with experimental findings. The Gibbs reaction energy (G) under the constant electrode potential can be expressed as (Ref. 25 in the revised Supplementary Information):

$$G = E_{\text{DFT}} + \text{ZPE} - \text{TS} - \mu_e \times N_e - \mu_{\text{cat/ani}} \times |N_e| \quad (\text{R5})$$

where μ_e is the chemical potential of the electron. N_e is the number of electrons added or removed in the system. $\mu_{\text{cat/ani}}$ is the chemical potential of the cations/anions. Moreover, the pH correction to the Gibbs free

energy can be given by (Ref. 16 in the revised Supplementary Information):

$$\Delta G_{pH} = -k_B T \ln(|H^*|) = -0.0592 pH \quad (R6)$$

For the Gibbs free energy difference (ΔG) between *OCHO and *COOH over different surfaces investigated in this work, ΔG can be calculated as:

$$\Delta G = (E_{DFT-^*COOH} - E_{DFT-^*OCHO}) + (ZPE_{^*COOH} - ZPE_{^*OCHO}) - T(S_{^*COOH} - S_{^*OCHO}) - \mu_e \times (N_{e-^*COOH} - N_{e-^*OCHO}) - \mu_{cat/ani} \times (|N_{e-^*COOH}| - |N_{e-^*OCHO}|) - 0.0592 pH \quad (R7)$$

The divergence in Gibbs free energy between the two pivotal intermediates across the specified surfaces are shown in Figure R6, and the corresponding configurations are presented in Figure R7. In the presence of as solvation, applied potential, and cation effects, our conclusions remain the same that the formation energy difference between *COOH and *OCHO species as a volcano plot across diverse surfaces. The Cu_6Sn_5 (-113) surface exhibits the most significant distinction between these critical intermediates with a thermodynamic favorability of forming *OCHO (the intermediate for FA). In other words, the Cu_6Sn_5 (-113) facet will show the highest selectivity in the formic acid formation process as compared to other examined catalysts.

Figure R6. Gibbs free energy difference between *COOH and *OCHO over Cu (111), $Cu_{0.86}Sn_{0.14}$ (111), Cu_6Sn_5 (-113) and Sn (100) surfaces in the presence of hybrid solvation effects and constant electrode potential of -2.1 V vs RHE at pH 1.

Figure R7. The adsorption configurations of two key intermediates (*OCHO and *COOH) on Cu (111), $\text{Cu}_{0.86}\text{Sn}_{0.14}$ (111), Cu_6Sn_5 (-113), and Sn (100) surfaces with considering hybrid solvation effects and utilizing CEP model with an applied potential of -2.1 V vs RHE.

To solve reviewer's concern, we have added this part on Page 10 in the manuscript:

Theoretical investigations were correspondingly carried out in this segment to investigate the effects of the electrochemical operating environment effects (e.g., electrolyte of 3 M K^+ , applied potential of $-2.1\text{ V}_{\text{RHE}}$, and $\text{pH} = 1$) on the selectivity for FA vs. CO formation using Constant Electrode Potential (CEP) model via performing Grand canonical DFT calculations.⁴⁵⁻⁴⁹ The selectivity for FA across Cu (111), $\text{Cu}_{0.86}\text{Sn}_{0.14}$ (111), Cu_6Sn_5 (-113), and Sn (100) surfaces exhibits a volcano plot (Fig. 2a and Supplementary Fig. 48), with Cu_6Sn_5 (-113) displaying the highest energy difference between the two pivotal intermediates, favoring the adsorption of *OCHO over CO. More details regarding grand canonical DFT simulations are provided in Supplementary Information.

On page 16:

To take the solvation effects into consideration, hybrid solvation effects have been employed in VASPsol along with explicit water molecules at the interface.⁴⁵⁻⁴⁸ The Constant Electrode Potential (CEP) model has been employed to investigate the constant negative applied potential and cation effects through grand-canonical DFT (GC-DFT) calculations.⁴⁹

The Supplementary Information part has been revised on Pages 9–11 regarding the electrolyte effects:

Hybrid solvation and applied potential effects. Here, we applied the hybrid solvation effects using constant electrode potential (CEP) model via performing grand-canonical DFT calculations (GC-DFT). We compared the formation energy of the two crucial intermediates (*OCHO and *COOH) to represent the selectivity of CO₂R to formic acid vs. CO. To account for solvation effects, we employed both the implicit solvation model incorporated in VASPsol, along with explicit water molecules at the interface.¹⁸⁻²¹ In the VASPsol model, the solvent was treated as a continuous medium, and its impact was represented using the dielectric constant of water. The electrolyte solution was modeled using a linearized Poisson-Boltzmann approach. Additionally, we implemented the Constant Electrode Potential (CEP) model to investigate surface charge and cation effects (e.g., K⁺) on the CO₂R reaction through GC-DFT calculations.²² At room temperature, the Debye screening length (Å) was calculated from the electrolyte concentration (M) using the following equation:²³

$$\kappa = \frac{3}{\sqrt{I}} \quad (18)$$

In the equation, κ represents the Debye length, and I denotes the electrolyte concentration (e.g., 3 M K⁺) in this part. In the CEP model, the applied potential was adjusted the Fermi level (E_f) towards a desired value by manipulating the number of electrons added or removed within the system. This adjustment by various number of the electrons can tune the work function (Φ) of the system and, thus, tune the applied potential (U_{SHE}) with a reference of the work function of Standard Hydrogen Electrode (Φ_{SHE}):

$$U_{SHE} = \frac{\Phi - \Phi_{SHE}}{e} \quad (19)$$

In the given context, Φ_{SHE} represents the work function of the standard hydrogen electrode, determined as $\Phi_{SHE} = 4.43$ eV using RPBE for the thermodynamic work function of the standard hydrogen electrode.²⁴ This value serves as a reference for adjusting the number of additional electrons introduced into the systems. By altering the number of extra electrons added into the systems, we ensure a consistent potential of ~ -2.1 V vs. RHE for each examined elementary reaction, aligning with experimental findings. The Gibbs reaction energy (G) under the constant electrode potential can be expressed as:²⁶

$$G = E_{DFT} + ZPE - TS - \mu_e \times N_e - \mu_{cat/ani} \times |N_e| \quad (20)$$

where μ_e is the chemical potential of the electron. N_e is the number of electrons added or removed in the

system. $\mu_{cat/ani}$ is the chemical potential of the cations/anions. Moreover, the pH correction to the Gibbs free energy can be given by:¹⁶

$$\Delta G_{pH} = -k_B T \ln([H^*]) = -0.0592 pH \quad (21)$$

For the Gibbs free energy difference (ΔG) between *OCHO and *COOH over different surfaces investigated in this work, ΔG can be calculated as:

$$\Delta G = (E_{DFT-*COOH} - E_{DFT-*OCHO}) + (ZPE_{*COOH} - ZPE_{*OCHO}) - T(S_{*COOH} - S_{*OCHO}) - \mu_e \times (N_{e-*COOH} - N_{e-*OCHO}) - \mu_{cat/ani} \times (|N_{e-*COOH}| - |N_{e-*OCHO}|) - 0.0592 pH \quad (22)$$

The divergence in Gibbs free energy between the two pivotal intermediates across the specified surfaces are shown in Fig. 2a in the manuscript, and the corresponding configurations are presented in Supplementary Fig. 48. In the presence of as solvation, applied potential, and cation effects, our conclusions remain the same that the formation energy difference between *COOH and *OCHO species as a volcano plot across diverse surfaces. The Cu_6Sn_5 (-113) surface exhibits the most significant distinction between these critical intermediates with a thermodynamic favorability of forming *OCHO (the intermediate for FA). In other words, the Cu_6Sn_5 (-113) facet will show the highest selectivity in the formic acid formation process as compared to other examined catalysts.

Fig. 2a Gibbs free energy difference between *COOH and *OCHO over Cu (111), $Cu_{0.86}Sn_{0.14}$ (111), Cu_6Sn_5 (-113) and Sn (100) surfaces in the presence of hybrid solvation effects and constant electrode potential of -2.1 V vs RHE at pH 1.

Supplementary Fig. 48 The adsorption configurations of two key intermediates (*OCHO and *COOH) on Cu (111), Cu_{0.86}Sn_{0.14} (111), Cu₆Sn₅ (-113), and Sn (100) surfaces with considering hybrid solvation effects and utilizing CEP model with an applied potential of -2.1 V vs RHE.

- 5) Since the electrolytic cell configuration substantially impacts CO₂R performance, testing consistency is needed. CO₂ performance was primarily evaluated in flow cells, whereas the *in-situ* ATR-FTIR tests were conducted in a three-electrode electrochemical single-cell. The rigor of the *in-situ* ATR-FTIR data is therefore questionable due to the mismatch in cell types.

Response: In light of the reviewer's comment, we performed additional studies to evaluate the CO₂R performance of Cu, Cu₆Sn₅, and Sn catalysts in an H-cell, aligning with the cell type used in the *in situ* attenuated total reflection Fourier transform infrared (ATR-FTIR) tests. (Please also note that, currently, there is a scarcity of flow cell devices suitable for *in situ* ATR-FTIR testing due to the limitations in collecting valid FTIR signals in the flow cell configuration.) Consistently, we employed the same electrolyte, 3 M KCl and 0.05 M H₂SO₄ at pH = 1, for the H-cell tests. As shown in Figure R7, Cu₆Sn₅ exhibits superior Faradaic

efficiencies (FEs) for formic acid production across all current densities ranging from 10 to 120 mA cm⁻² in the H-cell. This aligns with the results obtained in the *in situ* ATR-FTIR tests under the same operation conditions, which show an increased *OCHO coverage on Cu₆Sn₅. The performance in the H-cell is also consistent with the CO₂R results obtained in flow cells and membrane electrode assembly (MEA), reinforcing the significance of surface *OCHO coverage in promoting formic acid production.

Figure R7. Evaluation of the performance of Cu, Cu₆Sn₅, and Sn in H-cells using 3 M KCl and 0.05 M H₂SO₄ at pH = 1. The error bars presented data from three independent tests.

- 6) Further verification is needed to confirm the assignment of the *OCHO intermediate in the ATR-FTIR spectroscopy analysis.

Response: In light of the reviewer's comment, we performed *in situ* Raman analysis in a flow cell to examine the formic acid intermediate during CO₂R in the same electrolyte, 3 M KCl and 0.05 M H₂SO₄ at pH = 1. As shown in Figure R8, we observed vibrations at 1636 and 2810 cm⁻¹ corresponding to O–C–O and C–H in *OCHO, confirming the formation of HCOO⁻ during CO₂R. This Raman result aligns with those reported in previous *in situ* Raman studies for CO₂R (*J. Am. Chem. Soc.* 2020, **142**, (21), 9567–9581; *J. Chem. Phys.* 2019, **150**, 041718).

Furthermore, our *in situ* Raman studies revealed an increase in *OCHO peaks with the rise in applied negative potentials on Cu₆Sn₅ and Sn, indicating that Cu₆Sn₅ and Sn favor formic acid generation under negative potentials. Conversely, the *OCHO peaks decrease with increased negative potentials, consistent with the CO₂R performance observation that Cu produces C₂₊ rather than formic acid at higher negative potentials.

In addition, we confirmed that the peak at ~1384 cm⁻¹ in ATR-FTIR spectroscopy corresponds to the vibration peak of O–C–O, indicative of the *OCHO intermediate, which aligns with the results reported in *Nat. Nanotechnol.* 2021, **16**, 1386–1393; *Nat. Commun.* 2023, **14**, 2843. We have cited both references in the

Figure R8. *In situ* Raman spectra measured at different applied potentials for Cu_6Sn_5 (a), Sn (b), and Cu (c) under identical conditions (pH 1 in 3 M KCl and 0.05 M H_2SO_4 as the catholyte and 0.5 M H_2SO_4 as the anolyte) in a homemade Raman flow cell.

- 7) Infrared signal intensity depends on many factors. Direct comparison of peak area changes between different samples is insufficiently scientific. An internal standard methodology to benchmark relative peak areas may better verify variations in adsorption coverage. In addition, the approach for determining and comparing the peak areas in Fig. 4 is not explained. Details on the peak area calculation must be provided in the Methods.

Response: In light of the reviewer's recommendation, we conducted a series of new *in situ* ATR-FTIR tests, incorporating an internal standard of potassium ferricyanide in the electrolytes to facilitate peak area calibration. The selection of potassium ferricyanide as the internal standard was based on its distinct vibration peak at $\sim 2080 \text{ cm}^{-1}$, devoid of vibration peaks in the range of $1300\text{--}1400 \text{ cm}^{-1}$ that might overlap with the *OCHO intermediate peak during CO_2R .

We first examined the *in situ* ATR-FTIR during CO_2R with 5 mM potassium ferricyanide in 3 M KCl and 0.05 M H_2SO_4 , we observed a constant peak area of potassium ferricyanide within the potential range of -0.54 to $-0.94 \text{ V}_{\text{RHE}}$. However, at more negative potentials, the signal from potassium ferricyanide became unstable, rendering accurate calculation of the relative peak area of the *OCHO intermediate difficult. We, therefore, calculated the ratio (r_1) of the relative *OCHO peak areas on Cu_6Sn_5 and Sn at each potential within the

potential range of -0.54 to -0.94 V_{RHE} , using potassium ferricyanide as an internal standard (Table R1). For comparison, we also calculated the ratio (r_2) of the relative *OCHO peak areas on Cu_6Sn_5 and Sn at each potential within the same potential range of -0.54 to -0.94 V_{RHE} in electrolytes without potassium ferricyanide. Comparing r_1/r_2 , we found that the *OCHO peak areas on Cu_6Sn_5 in electrolytes without potassium ferricyanide in Figure 3 in the originally submitted manuscript should be 1.4 times higher following calibration using the calibrated ratio (r_1). Correspondingly, the plateau *OCHO peak intensity on Cu_6Sn_5 measured at potentials above -1.04 V_{RHE} in the CO_2R electrolyte without potassium ferricyanide is calculated to be ~ 2.8 times higher than that of Sn under the same operating conditions in electrolyte without potassium ferricyanide (Figure R9).

It is important to note that direct comparison of the relative *OCHO peak areas on Cu_6Sn_5 and Sn at high operating potentials above -1.04 V_{RHE} in the CO_2R electrolyte with the internal standard of potassium ferricyanide may be inappropriate. This is because potassium ferricyanide is determined to be unstable at potentials above -1.04 V_{RHE} , and it is unclear whether the instability may influence the *OCHO surface coverage. Therefore, our established calibration using the relative peak ratios allows for a better comparison of the relative surface *OCHO peak areas in the *in situ* CO_2R ATR-FTIR measurements.

In addressing other comments from the reviewer, we have provided the calculation details of peak areas and explanation on comparing peak intensities on page 18 in the Methods section of the revised manuscript and on page 5 in the revised Supplementary Information.

Table R1. The relative peak areas of *OCHO intermediates for Cu_6Sn_5 , Cu, and Sn catalysts at the same potential range of -0.54 to -0.94 V_{RHE} . The calibrated *OCHO ratio on Cu_6Sn_5 and Sn (r_1/r_2) is 1.37 ± 0.5 .

Potential (V_{RHE})	*OCHO peak area on Cu_6Sn_5 calibrated using the peak areas of potassium ferricyanide as an internal standard	*OCHO peak area on Sn calibrated using the peak areas of potassium ferricyanide as an internal standard	*OCHO ratio on Cu_6Sn_5 and Sn with internal standard (r_1)	*OCHO peak area on Cu_6Sn_5 without using internal standard	*OCHO peak area on Sn without using internal standard	*OCHO ratio on Cu_6Sn_5 and Sn without internal standard (r_2)	r_1/r_2
-0.54	84.6	9.4	9.00	0.4692	0.0667	7.03	1.28
-0.64	89.3	16.5	5.41	0.5429	0.1477	3.68	1.47

-0.74	87.7	28.3	3.10	0.7188	0.3225	2.23	1.39
-0.84	94.8	40.7	2.33	1.2799	0.7348	1.74	1.34
-0.94	86.8	56.1	1.55	2.7497	2.6063	1.06	1.47

Figure R9. (a–c) The ATR-FITR results in the potential range of -0.54 to $-0.94 \text{ V}_{\text{RHE}}$, which we used to calculate the integrated FTIR peak areas. (d) The integrated FTIR peak areas of Cu_6Sn_5 . (e) The calibrated *OCHO peak areas on Cu_6Sn_5 .

- 8) To identify the active sites, it might be important to characterize the catalyst post- CO_2R more rigorously, such as morphology, oxidation states, and other physical and chemical information of the catalysts after stability testing.

Response: To elaborate on the reviewer's recommendation, we characterized the morphology, oxidation states, and crystallinity of the Cu_6Sn_5 catalyst after the 120-hour stability test under strong acid conditions. The SEM images in Figure R10 reveal the morphology of Cu_6Sn_5 remains well-preserved before and after the reaction. The crystallinity analysis, from XRD and HRTEM, confirmed the persistence of the Cu_6Sn_5 phase (Figures R2 and R3). Surface chemical states were characterized by XPS, showing Sn^{4+} in combination with Sn^0 , as well as mixed Cu^0 or Cu^{1+} after the 120-h CO_2R reaction (Figure R11). Please also note that the oxidation states of Cu and Sn could be post oxidation after removal of the catalysts from the electrochemical cells.

Figure R2. Comparison of X-ray diffraction (XRD) patterns for Cu_6Sn_5 before and after the 120-hour stability test at 0.5 A cm^{-2} under both pH 1 and pH 14 conditions.

Figure R3. (a) Scanning electron microscopy (SEM), (b) high-resolution transmission electron microscopy (HRTEM), and (c-h) scanning transmission electron microscopy with energy dispersive X-ray spectroscopy (STEM-EDX) images of Cu_6Sn_5 following the 120-hour stability test conducted at 0.5 A cm^{-2} at pH 1.

Figure R10. The SEM images of the Cu_6Sn_5 catalyst before (a) and after (b) the 120-hour stability test under strong acid conditions.

Figure R11. XPS spectra of Cu_6Sn_5 catalyst ((a) Cu 2p; (b) Sn 3d) after the 120-hour stability test under strong acid conditions.

- 9) The performance under different acidic conditions needs to be investigated, especially stronger acids like 0.1 M or 1 M H_2SO_4 . Moreover, the practical applications of CO_2 reduction are hindered by the high concentration of K^+ ions.

Response: In light of the reviewer's suggestion, we returned to the lab and conducted a series of new experiments with Cu_6Sn_5 under different acidic electrolytes, replacing 0.05 M H_2SO_4 with stronger acids such as 0.1 and 1 M H_2SO_4 , with the presence of 3 M KCl in electrolytes. Performance tests in flow cells using Cu_6Sn_5 (as depicted in Figure R12) revealed that as the pH decreases, HER increases at small current densities. At a current density of 500 mA cm^{-2} , formic acid production from CO_2R in all acids becomes dominant, because high current density substantially consume protons near the catalyst surfaces, leading to an increase in local pH. Therefore, under high current density conditions, HER is weaker compared to low current density conditions.

To produce pure formic acid without K^+ , we used a K^+ -free, solid-state electrolyte (SSE)-based MEA system and different anolytes (0.1, 0.5, and 1 M H_2SO_4). As shown in Figure R13, we found that Cu_6Sn_5 performs the best in the 0.5 M H_2SO_4 anolyte with a notable formic acid Faradaic efficiency of above 95%. We have added the related discussions on page 15 in the revised manuscript and in Figure S54 in the revised Supplementary Information.

Figure R12. The CO₂R performance of Cu₆Sn₅ in different catholytes of 0.1 and 1 M H₂SO₄ with 3 M KCl at different applied current densities in a flow cell.

Figure R13. The CO₂R performance of Cu₆Sn₅ in different analytes of 0.1, 0.5 and 1 M H₂SO₄ at different applied current densities in an SSE-based MEA.

- 10) Improving the single-pass CO₂ utilization by reducing the CO₂ flow rate lacks scientific significance, since reducing the CO₂ flow rate may not actually enhance the catalytic activity. Providing the performance of CO₂R dependent on current density at different CO₂ flow rates is essential.

Response: In light of the reviewer's comment, we conducted additional SPCE studies, varying flow rates (3, 5, 10, 15, 18.5 sccm) and current densities (100, 300, 500 mA cm⁻²). The results are presented in Figure R14.

Figure R14. Single-pass carbon efficiency (SPCE) of CO₂R on Cu₆Sn₅ at different current densities of 100, 300, and 500 mA cm⁻², tested in acidic and alkaline electrolytes with varying CO₂ flow rates.

The SPCE was calculated using the formula (R8):

$$SPCE = \frac{(I_{HCOOH} \times 60 s) / (N \times F)}{(v \times 1 \text{ min}) / (24.05 \text{ L mol}^{-1})} = \frac{n_{HCOOH}}{n_{CO_2}} \quad (R8)$$

Where I_{HCOOH} represents the partial current of HCOOH in amperes, N stands for the electron transfer, which is 2 for HCOOH, F is Faraday's constant, which is 96485 C mol⁻¹, v is the flow rate of CO₂ in L min⁻¹.

We have included relevant discussion on page 4 in the revised Supplementary Information.

Reviewer #2 (Remarks to the Author):

The authors present a CuSn catalyst for high-rate CO₂ electrolysis in alkaline and acidic conditions, as well as in a solid-state electrolyte MEA to produce formic acid at competitive reaction rates. The selectivity was linked to the coverage of a key formic acid intermediate alongside weakened *H binding energy. Overall, the conclusions are well-supported and the work provides a significant step in the development of CO₂ electrolyzers. However, there is some key information missing and clarification on specific points required as mentioned below. Should the authors address these comments, I recommend publication of this manuscript in Nature Communications.

Response: We express our gratitude to the reviewer for approving our work and providing valuable comments that have enabled us to improve the quality of our work. In light of the advice, we have performed new experiments to address all concerns, as documented in the point-by-point responses below. We have also highlighted the changes in yellow in the revised manuscript.

- 1) It is not clear how the overpotential for formic acid production was calculated. Is this the overpotential for a given current density, or the onset potential, in which case the generation of formic acid would need to be confirmed as a sole product. Can the authors please verify how this was assessed.

Response: Thank you for the comment. In our work, the overpotential refers to the additional potentials

needed to drive the CO₂R to formic acid reaction at certain rates (i.e. overpotential of 1.2 V at 96 mA cm⁻² for formic acid generation). This definition is presented in *ACS Catal.* 2014, **4**, 630. We have enhanced the clarity when discussing the overpotential on page 18 in the Methods section of the revised manuscript and on page 4 in the revised Supplementary Information.

Following the reviewer's suggestion, we plotted the partial current density of formic acid as a function of the applied potential in Figure R15. The current density in this Figure exclusively pertains to formic acid generation, and the corresponding overpotential (E_{op}) for formic acid generation was estimated using the equation (R9):

$$E_{op} = E_{rp} - E_{tp} \quad (R9)$$

Where E_{rp} is the applied potential, and E_{tp} is the theoretical potential for formic acid generation from CO₂. From Figure R13, we observed that among all the Cu_{1-x}Sn_x catalysts, Cu_{0.56}Sn_{0.44} (Cu₆Sn₅) shows the lowest potentials needed for formate production (Fig. 2c in the revised manuscript). We have revised this sentence on page 10 in the revised manuscript.

Figure R15. The partial current densities of formic acid as a function of applied potentials on Cu, Cu_{1-x}Sn_x ($x = 0.08, 0.14, 0.28, 0.44, 0.71, 0.88$), and Sn catalysts in the acidic electrolyte of 3 M KCl and 0.05 M H₂SO₄ at pH 1.

- 2) No data is provided for the other CuSn catalyst compositions which could be responsible for selectivity changes. It would be good to see this data to verify that the catalyst chosen was indeed the best-performing.

Response: In light of the reviewer's suggestion, we evaluated the CO₂R performance for Cu_{1-x}Sn_x ($x = 0.08, 0.14, 0.28, 0.44, 0.71, 0.88$) catalysts in 3 M KCl and 0.05 M H₂SO₄ at pH = 1. As shown in Figure R16, Cu₆Sn₅ exhibits the highest selectivity for formic acid production under acidic conditions. This result aligns with the results obtained from CO₂ electrolysis under alkaline conditions. We have incorporated these results in Figure 2 in the revised manuscript. The relevant discussions are also presented on page 10 in the revised manuscript.

Figure R16. Faradaic efficiencies (FEs) and energy efficiencies (EEs) for Cu, Cu_{1-x}Sn_x ($x = 0.08, 0.14, 0.28, 0.44, 0.71, 0.88$) and Sn catalysts at various current densities of 0.1, 0.4, 0.8, 1.2, and 1.6 A cm⁻² in 3 M KCl and 0.05 M H₂SO₄ at pH = 1 in flow cells. The error bars presented are derived from three independent tests.

- 3) Information about the SPCE is missing. How long was each experiment to get a single point on the SPCE plot conducted for? What equations were used for SPCE calculations?

Response: Thank you for the comment. We have enhanced the clarity of the experimental details for measuring SPCE and the associated information for SPCE calculation on page 18 in the Methods section in the revised manuscript and on page 4 in the revised Supplementary Information.

To elaborate, SPCE measurements were conducted in an air-tight flow cell setup (homemade) at varying CO₂ flow rates. The experimental conditions were specified as follows:

- For measuring SPCE under acidic conditions: we used 3 M KCl and 0.05 M H₂SO₄ at pH = 1 as the catholyte, and the anolytes were 0.5 M H₂SO₄.
- For measuring SPCE under conditions referred to as "neutral" in our previous submission: the catholyte was 3 M KCl solution at pH = 6.2, while the anolyte was 0.5 M H₂SO₄ at pH = 1. The use of different electrolyte pH for CO₂R were first reported in *Science* 2020, **367**, 661–666. Using an acidic anolyte is expected to reduce the operating potentials on anodes when using IrO_x as the OER catalyst. To confirm the result, we re-conducted the SPCE experiments under the same operating conditions (Figure R17). We also record the pH of the catholyte during different time course of the SPEC measurement. We found that the catholyte pH gradually decreases to below 3 after 20-min operation (Figure R18). Therefore, the SPCE we obtained under this operating condition resembles that obtained in acidic conditions. We appreciate this comment and have revised the corresponding contents on pages 10–11 in the manuscript.
- For measuring SPCE under alkaline conditions: the catholyte and anolyte were 1 M KOH solution at pH = 14.

The SPCE results were calculated based on a 1-hour electrolysis, with selectivity measurements for formic

acid taken after reactions. The calculation formula is:

$$SPCE = \frac{(I_{HCOOH} \times 60 \text{ s}) / (N \times F)}{(v \times 1 \text{ min}) / (24.05 \text{ L mol}^{-1})} = \frac{n_{HCOOH}}{n_{CO_2}} \quad (R8)$$

Where I_{HCOOH} represents the partial current of HCOOH in amperes, N stands for the electron transfer, which is 2 for HCOOH, F is Faraday's constant, which is 96485 C mol^{-1} , v is the flow rate of CO_2 in $L \text{ min}^{-1}$.

Figure R17. SPCE of CO_2R on Cu_6Sn_5 at 0.5 A cm^{-2} obtained in acidic (pH 1), "neutral" (pH 2.4 measured after 20-min operation using $0.5 \text{ M H}_2\text{SO}_4$ electrolyte at $\text{pH} = 1$), and alkaline (pH 14) electrolytes at different CO_2 flow rates of 3, 6, 10, 15, 18.5 sccm.

- 4) The authors mention that in $\text{pH} < 1$ solutions carbonate formation is rare, however recent reports rather show that it occurs close to the electrode but can be reconverted back to CO_2 . The SPCE plots suggest that at pH 6.2, there is limited consumption of CO_2 by hydroxide to form carbonate. However, as the local pH increases under electrolysis conditions, the plots should display similar features to the alkaline feed. Can the authors explain this trend and account for the low loss of CO_2 to carbonate here.

Response: Thank you for the comment. To elaborate, SPCE measurements were conducted in an air-tight flow cell setup (homemade) at varying CO_2 flow rates. The experimental conditions were specified as follows:

- For measuring SPCE under acidic conditions: we used 1 M KCl and $0.05 \text{ M H}_2\text{SO}_4$ at $\text{pH} = 1$ as the catholyte, and the anolytes were $0.5 \text{ M H}_2\text{SO}_4$.
- For measuring SPCE under neutral conditions: the cathode side used 3 M KCl solution at $\text{pH} = 6.2$, while the anode side used $0.5 \text{ M H}_2\text{SO}_4$ at $\text{pH} = 1$. These conditions were reported in ref. *Science* 2020, **367**, 661–666 in which the pH of anolyte and cathode is different. Using an acidic anolyte is expected to reduce the overpotential on the anode side when using IrO_x as the OER catalyst; however, we found that this configuration gradually decreases the pH of the catholyte to below 3 during 20-min operation (Figure R18).

Therefore, the obtained SPCE obtained under this condition is similar to that obtained in the acidic condition. We appreciate the comment and have therefore revised the corresponding contents in the manuscript and Supplementary Information.

- For measuring SPCE under alkaline conditions: the catholyte and anolyte were 1 M KOH solution at pH = 14.

We have revised the corresponding contents on pages 10–11 in the manuscript considering that the focus of this work is the acidic CO₂R using Cu-Sn based electrocatalysts.

Figure R17. SPCE of CO₂R on Cu₆Sn₅ at 0.5 A cm⁻², obtained in acidic (pH 1), "neutral" (pH 2.4 measured after 20-min operation using 0.5 M H₂SO₄ electrolyte at pH = 1), and alkaline (pH 14) electrolytes at different CO₂ flow rates of 3, 6, 10, 15, 18.5 sccm.

Figure R18. The pH of catholyte (3 M KCl) changes over time during the SPCE test and reaches 2.4 after 20-min operation.

- 5) Although XPS data is mentioned in the methods, no results are provided. There are required to assess the degree of electronic modulation that occurs upon alteration of the Sn/Cu ratio, as this data can also be correlated to selectivity changes due to the modulation of the d-band.

Response: We conducted the XPS tests on $\text{Cu}_{1-x}\text{Sn}_x$ ($x = 0.08, 0.14, 0.28, 0.44, 0.71, 0.88$) catalysts (Figure R19), and found that the strong peaks at 494.4 and 486.0 eV corresponded to $\text{Sn}^{4+} 3d_{3/2}$ and $\text{Sn}^{4+} 3d_{5/2}$ and the peaks at 493.0 eV, 484.6 eV corresponded to $\text{Sn}^0 3d_{3/2}$ and $\text{Sn}^0 3d_{5/2}$. From the high-resolution XPS spectrum of Cu 2p, the peaks at 932.0 and 951.8 eV corresponded to $\text{Cu}^{0,1+}$, while the peaks at 934.2 and 953.9 eV were consistent with Cu^{2+} . The positively charged Sn^{4+} and Cu^{2+} are detected, probably because of the surface oxidation that occurred during the storage of the sample in air. However, the electronic modulation in the valence band spectrum was not clearly observed when we altered the Sn/Cu ratio (Figure R20). Each of the XPS work function analysis was conducted after a 5-min. Ar bombardment treatment to get rid of the surface oxidation layer on $\text{Cu}_{1-x}\text{Sn}_x$. We suggest that electronic band structure could be determined by many factors such as metal facets, metal concentrations, etc., and in our case, it is not clear of the modulation of the d-band of Cu to the final HCOOH performance.

Figure R19. The XPS spectra of $\text{Cu}_{1-x}\text{Sn}_x$ ($x = 0.08, 0.14, 0.28, 0.44, 0.71, 0.88$) catalysts.

Figure R20. Surface valence band photoemission spectra of Cu, $\text{Cu}_{1-x}\text{Sn}_x$ ($x = 0.08, 0.14, 0.28, 0.44, 0.71, 0.88$) and Sn. The white bar in highlights the d-band center of various materials.

- 6) Electrolysis duration is not provided in most cases for data presented apart from for the long-term data. This key information is important to help verify that setups are not due to transient effects. This is particularly important for the SPCE plots.

Response: Thank you for the comment. We have improved the clarity of our manuscript on page 18 in the revised manuscript: "All data, including Faradaic efficiencies, were collected based on 1-hour electrolysis. Stability tests were conducted over 300 hours." Therefore, our presented data is not a result of transient effects. We have also included the related key information of the stability curve for measuring SPCE in the revised Supplementary Information.

- 7) Several studies are missing from figure S41 and the main text that show high FE for formate and should be added: <https://doi.org/10.1002/advs.201902989>; <https://doi.org/10.1002/anie.202206279>; <https://doi.org/10.1002/anie.202110000>; <https://doi.org/10.1002/adma.202002822>; DOI: 10.1002/adfm.202213145

Response: Thank you for the comment. We have cited these references in the reference section in the revised

manuscript and on pages 57 and 69–70 in comparison Figure S46 and Table S2 in the revised Supplementary Information.

- 8) The authors should provide potentials for a range of current densities to facilitate comparison of energy efficiency with other work.

Response: In light of the reviewer's recommendation, we have analyzed Cu, Sn, and Cu₆Sn₅ catalysts under both acidic and alkaline conditions, measuring Faradaic efficiencies at various current densities. As shown in Figures R16 and R21, it can be found that the Cu₆Sn₅ catalyst exhibits higher energy efficiency than Sn and Cu at different current densities.

Figure R16. Faradaic efficiencies (FEs) and energy efficiencies (EEs) for Cu, Cu_{1-x}Sn_x ($x = 0.08, 0.14, 0.28, 0.44, 0.71, 0.88$) and Sn catalysts at various current densities of 0.1, 0.4, 0.8, 1.2, and 1.6 A cm⁻² in 3 M KCl and 0.05 M H₂SO₄ at pH = 1 in flow cells. The error bars presented are derived from three independent tests.

Figure R21. Faradaic efficiencies (FEs) and energy efficiencies (EEs) for Cu, Cu_{1-x}Sn_x ($x = 0.08, 0.14, 0.28, 0.44, 0.71, 0.88$) and Sn catalysts at various current densities of 0.4, 0.8, 1.2, 1.6, and 2.0 A cm⁻² in 1 M KOH at pH = 14 in flow cells. The error bars presented are derived from three independent tests.

Upon comparing our findings with existing literature, we propose that Cu₆Sn₅ achieves one of the best energy efficiencies under both alkaline and acidic conditions (Table R2).

Table R2. Comparison of our work with previously published data under both acidic and alkaline conditions.

Cat.	electrolyte	FE (%)	EE (%)	Stability (h)	Ref.
Cu_6Sn_5	3 M KCl +0.05 M H_2SO_4	96	52.2	300	This work
Cu_6Sn_5	1 M KOH	92	57.9	120	This work
Cu_6Sn_5	MEA	88	37	130	This work
sulfur-doped indium	0.5 M KHCO_3	93	–	–	6
surface-lithium-doped tin	1 M KOH	92	–	150	11
SiC- Nafion TM /SnBi/PTFE	MEA	>90	–	125	12
Sn(S)-H	0.5 M K_2SO_4 + H_2SO_4 (pH = 3)	92.1	–	14	15
2D-Bi	MEA	>90	48.5/43.3	100	19
Bi nanosheets	3 M KCl +0.05 M H_2SO_4	92.2	–	8	20
3D Sn/CNT-Agls	0.5 M KHCO_3	82.7	–	–	26
SnO_2 nanoparticles	1 M KHCO_3	64	–	–	27
nBuLi-Bi	MEA	~97	–	100	28
InP CQDs	1 M KOH	93	–	4	29
Sn quantum sheets	0.1 M NaHCO_3	89	–	50	30
BiOBr-templated catalyst	1 M KHCO_3	>90	–	65	31
carbon-supported SnO_2	MEA	90	15.3	11	32
carbon-supported SnO_2	0.1 M KHCO_3 / 1 M KOH	75	10–41	3.5/3	33
SnO_2 nanoparticles	5 M KOH	70	–	–	34
Sn nanoparticles	0.45 M KHCO_3 +0.5 M KCl	70	–	–	35
Sn/ SnO_2 nanoparticles	0.5 M Na_2CO_3 +0.5 M Na_2SO_4	>70	26.11–68. 06	–	36
Bi/ Bi_2O_3 nanosheets	0.5 M KHCO_3	~90	>50	31	37
Sn/Cu HFGDE	0.5 M KHCO_3	78	–	4	38
Pb_1Cu	MEA	85	35.2	180	39
Bi@Sn core-shell nanoparticles	2 M KHCO_3	92	56	20	40
Cu-SPy	5 M KOH	81	31	6	41
Cu_2SnS_3 nanosheets	0.5 M KHCO_3	83.4	–	–	42
Bi- SnO_x nanoshells	0.5 M KHCO_3	95.8	–	50	43

sulfur-modified Cu ₂ O(S3-Cu ₂ O-70)	0.1 M KHCO ₃	~90	–	83	44
---	-------------------------	-----	---	----	----

- 9) Line 113 – the reference to Figure 1a is not described in detail. The authors should provide a clear explanation to the altered pathways available when heteroatoms are present or simplify the schematic to clearly show the different pathways.

Response: Thank you for the comments. We have now explained more clearly for Figure 1a on page 5, lines 107–114 in the revised manuscript "Copper (Cu) has established itself as a predominant and cost-effective electrocatalyst capable of generating a variety of hydrocarbons through CO₂R. Previous research has revealed effective strategies for modifying the *OCHO-binding and *CO-binding properties on surfaces of Cu-based alloy surfaces by introducing foreign elements (e.g., Zn, Al, Pb) into the Cu lattice. Specifically, the inclusion of 5–10 at.% Zn or Al into Cu has shown the ability to partially weaken *CO adsorption on the Al or Zn modified Cu site compared to the adjacent Cu-Cu site, thereby creating asymmetric *CO binding energies for improved C₂₊ production. (*Nature* 2020, **581**, 178–183; *Nat. Commun.* 2023, **14**, 1298). Additionally, the introduction of a single Pb atom into Cu has been observed to enhance selectivity towards formic acid. (*Nat. Nanotechnol.* 2021, **16**, 1386–1393). These alternations selectively promote the production of FA or CO/C₂₊ via distinctive reaction pathways (Fig. 1a)".

- 10) In Fig S45, what was the flow rate of CO₂ in this experiment and the corresponding SPCE?

Response: In Figure S45 (revised Figure S47), the CO₂ flow rate is set at 25 standard cubic centimeter per minute (sccm), resulting in an SPCE of 2.6% (0.125 A, 0.25 cm², 70% FE, 25 sccm). To evaluate the catalyst's stability for CO₂R, we intentionally maintain a sufficiently high CO₂ flow rate to mitigate potential instability factors arising from insufficient CO₂.

- 11) Line 268 – I could not find the additional crystallographic data mentioned.

Response: Thank you for the comment. We have incorporated the crystallographic data on pag 61–64 in Figures S50–S53.

REVIEWERS' COMMENTS

Reviewer #1 (Remarks to the Author):

The authors have revised the manuscript carefully on the basis of the reviewers, and I think the revised manuscript is acceptable.

Reviewer #2 (Remarks to the Author):

The authors have provided full answers to all queries and I have no additional comments. Therefore I recommend publication of this revised manuscript.

Response to Reviewers

Reviewer #1

- The authors have revised the manuscript carefully on the basis of the reviewers, and I think the revised manuscript is acceptable.

Response: We thank the reviewer for the positive comments.

Reviewer #2

- The authors have provided full answers to all queries and I have no additional comments. Therefore I recommend publication of this revised manuscript.

Response: We thank the reviewer for the positive comments.